# CHD8 interacts with BCL11A to induce oncogenic transcription in triple negative breast cancer

Mark Waterhouse[1,2,7], Kyren Lazarus [1,7], Maria Francesca Santolla [1,3,6,7], Sara Pensa[1,2], Eleanor Williams [2], Abigail J Q Siu [1,2], Hisham Mohammed[4], Irina Mohorianu[2], Marcello Maggiolini[3], Jason Carroll [5], Laura S Itzhaki[1], Taufiq Rahman [1] & Walid T Khaled [1,2✉]

## Abstract

**The identification of tumour-specific protein–protein interactions remains a challenge for the development of targeted cancer therapies. In this study we describe our approach for the identification of triple negative breast cancer (TNBC)-specific protein–protein interactions focusing on the oncogene BCL11A. We used a proteomic approach to identify the BCL11A protein networks in TNBC and compared it to its network in B-cells, a cell type in which BCL11A plays crucial roles. This approach identified the chromatin remodeller CHD8 as a TNBC-specific interaction partner of BCL11A. We show that CHD8 also plays a key role in TNBC pathogenesis, with detailed multi-omics analysis revealing that BCL11A and CHD8 co-regulate several targets and synergise to drive tumour development and progression. Using a battery of biophysical assays, we confirm that the BCL11A-CHD8 interaction is direct and identify chemical fragments that disrupt this interaction and affect downstream targets, decreasing proliferation in 3D colony assays. Our study provides a proof-of-principle approach for investigating tumour-specific protein–protein interactions and identifies lead chemical compounds that could be developed into novel therapeutics for TNBC.**

**Keywords** TNBC; BCL11A; CHD8; Tumour Specific Protein–protein Interactions

**Subject Categories** Cancer; Chromatin, Transcription & Genomics; Pharmacology & Drug Discovery

## Introduction

One of the major challenges in oncology is the development of drugs that are specific to and effective in killing tumour cells without eliciting undesirable side effects on healthy tissue. Advances in genomics over the past 2 decades has facilitated significant leaps in identifying gene expression (Koboldt et al, 2012), mutations (Davies et al, 2002), genomic alterations (Capdeville et al, 2002) and synthetic lethality vulnerabilities (Chan and Giaccia, 2011) that define the myriad of different tumour types. These approaches have led to the development of novel and successful targeted therapies. One area of biology that had remained underexplored for the development of targeted therapeutics is the tumour specific protein–protein interaction (TSPPI) space (Ivanov et al, 2013; Sharifi Tabar et al, 2022). This approach could be particularly useful in targeting unmutated proteins that play a key role in the pathology of the disease. In this study we investigate novel TSPPIs in the context of Triple Negative Breast Cancer (TNBC).

TNBC is an aggressive form of breast cancer that accounts for 15–20% of all breast cancer cases (Carey et al, 2010). TNBC patients are at higher risk of developing secondary and metastatic disease and hence have lower rates of survival. Identifying genes that are responsible for the development of TNBC is therefore essential for the development of targeted therapies. We have previously reported the finding of one such gene, *BCL11A* (Khaled et al, 2015), which is overexpressed in the majority of TNBCs and promotes tumour formation making it an attractive target for the treatment of TNBC tumours. Despite its key role in driving tumour development no significant mutations have been reported in *BCL11A*, suggesting a strong selection pressure to keep it intact. This, however, presents a problem for the development of targeted therapies against BCL11A as BCL11A is a regulatory C2H2 type zinc finger transcription factor that plays key roles in other tissues including the brain and hematopoietic system (Yin et al, 2019). Therefore, the development of therapies that can achieve specificity in the target tissues of interest is crucial.

To address these challenges, here we report the identification of a BCL11A TSPPI with the chromatin remodeller CHD8 in TNBC. Crucially, the complex was detected in TNBC but not in other breast cancer subtypes or primary mouse B-cells. Through a battery of biophysics and multiomic analyses, we show that BCL11A and CHD8 directly interact to create a functional complex that drives an oncogenic transcriptional programme. Further to this, we also describe the identification of several hits from an in silico

---

[1]Department of Pharmacology, University of Cambridge, CB2 1PD Cambridge, UK. [2]Cambridge Stem Cell Institute, University of Cambridge, CB2 0AW Cambridge, UK. [3]Department of Pharmacy, Health and Nutritional Sciences, University of Calabria, 87036Rende, Cosenza, Italy. [4]Cancer Early Detection Advanced Research Center, Knight Cancer Institute, Oregon Health & Science University, Portland, OR 97201, USA. [5]CRUK, Cambridge Institute, University of Cambridge, CB2 0AN Cambridge, UK. [6]Present address: Department of Life Sciences, Health, and Healthcare Professions, Link Campus University, 00165 Rome, Italy. [7]These authors contributed equally: Mark Waterhouse, Kyren Lazarus, Maria Francesca Santolla. ✉E-mail: wtk22@cam.ac.uk

fragment-based drug discovery (FBDD), and demonstrate their ability to prevent BCL11A-CHD8 complex formation using in vitro biophysical and functional assays.

## Results

### Identification of BCL11A TSPPIs in TNBC

To identify BCL11A TSPPIs we used Rapid Immunoprecipitation and Mass Spectrometry of Endogenous Proteins (RIME) (Mohammed et al, 2016, 2013) on a range of TNBC samples (MDA-MB-231, 4T1, PDX) as well as primary B-cells derived from C57BL/6 mice (Fig. 1A). In total, 12 samples were analysed by RIME. This identified 1591 proteins that interact with BCL11A in TNBC ($n = 8$) and 580 proteins that interact with Bcl11a in primary mouse B-cells ($n = 2$). 139 proteins were shared between the TNBC and B-cell RIME experiments, resulting in 1452 proteins that interact with BCL11A in TNBC only (Fig. 1A, Dataset EV1–6). CHD8, a chromatin remodelling factor, was the only protein identified in all TNBC samples but not in B-cells. CHD8 and BCL11A are also both expressed in the brain and play important roles in autism spectrum disorders (Yin et al, 2019). Accordingly, we performed a BCL11A RIME experiment using wild-type female mouse brain tissue to investigate whether these proteins interact in the brain (Dataset EV14). This resulted in the pull-down of BCL11A but not CHD8, supporting the specific nature of the BCL11A-CHD8 interaction in TNBC.

Due to the high reproducibility and specificity of CHD8 pulldown by BCL11A, we then investigated whether CHD8 also plays a role in TNBC using either shRNA- or CRISPR-mediated *Chd8* knockdown or deletion in 4T1 cells (Fig. 1B). This showed that knockdown of *Chd8* using either method reduces the ability of these cells to form tumours both in vitro (3D colony formation assays) and in vivo (Xenograft tumours). Similar investigations using the human MDA-MB-231 TNBC cell line also gave significant reductions in colony formation in a 3D-colony assay (Appendix Fig. S1A–C), thus phenocopying BCL11A and suggesting a key role for CHD8 in TNBC

To further explore the expression of CHD8 in TNBC we analysed PDX samples which identified that, at the protein level, BCL11A and CHD8 are specifically upregulated in TNBC PDXs but not in Luminal breast cancer PDXs (Fig. 1C). Similarly, Bcl11a and Chd8 protein levels were also elevated in tumours derived from the mouse TNBC model (*Brca1^{f/f} p53^{+/-} Blg-Cre*) (Fig. 1D). Surprisingly, in both cases, whereas BCL11A was upregulated at both the protein and RNA level, CHD8 appeared to be upregulated at the protein level only (Fig. 1C,D). Additional analysis of *BCL11A* and *CHD8* mRNA expression across the METABRIC and TCGA cancer datasets confirmed this observation and showed that *BCL11A* is upregulated in basal (TNBC) breast tumour samples compared to luminal breast tumours, whereas *CHD8* shows no differences in expression across different breast tumours (Appendix Fig. S1E). There was also no correlation between *BCL11A* and *CHD8* expression at the mRNA level. Given the simultaneous upregulation of CHD8 with BCL11A at the protein level, we then sought to understand whether BCL11A influences CHD8 protein levels. Using shRNA-mediated knockdown of *Bcl11a* in 4T1 cells, we showed that knockdown of *Bcl11a* leads to a simultaneous

reduction of CHD8 at the protein level but not at the RNA level (Fig. 1E). Taken together, this data suggests that BCL11A may facilitate or stabilise CHD8 expression in this context and together they drive a TNBC specific transcription programme.

### BCL11A and CHD8 co-localise on the genome and regulate target genes

To investigate the hypothesis that BCL11A and CHD8 cooperate to drive a TNBC-specific transcription programme, we performed multiomic analyses comprising RNAseq, ChIPseq and ATACseq on 4T1 cells transfected with control shRNA, *Bcl11a*-shRNA, or *Chd8*-shRNA (Fig. 2A; Appendix Fig. S1). We first considered the RNAseq dataset and performed differential expression analysis of knockdown cells compared to control cells. This identified 624 genes that were affected by *Bcl11a* knockdown and 820 genes that were affected by *Chd8* knockdown (Appendix Fig. S2A, Dataset EV7). Of the 624 genes affected by *Bcl11a* knockdown, 395 were found to be increased upon knockdown whereas 229 were found to be decreased upon knockdown. Of the 820 genes affected by *Chd8* knockdown, 504 were found to be increased and 316 were found to be decreased upon knockdown (Fig. 2B; Appendix Fig. S2B,C).

Comparison of the genes affected by *Bcl11a* or *Chd8* knockdown identified a significant overlap in target genes, with 368 genes affected by knockdown of either gene (Fig. 2B; Appendix Fig. S2A). Of these genes, 237 were found to be negatively regulated by *Bcl11a* or *Chd8* knockdown (Fig. 2B; Appendix Fig. S2B Quadrant A), whereas 130 genes were found to be positively regulated (Fig. 2B; Appendix Fig. S2B Quadrant C). One gene (*Ube2ql1*) was found to be negatively regulated by Bcl11a but positively regulated by Chd8 (Appendix Fig. S2B Quadrant B). The top 30 upregulated and downregulated genes can be seen in Fig. 2C. Linear regression and trendline analysis identified an $R^2$ value of 0.8983, suggesting a strong correlation of *Bcl11a* or *Chd8* knockdown on this shared gene set and suggesting a potential co-regulatory role for these proteins (Appendix Fig. S2A). Gene set enrichment analysis (GSEA) on each shared gene quadrant using g:Profiler identified changes in various pathways including cell cycle and DNA repair pathways (Appendix Fig. S3).

We then investigated the genomic binding properties of these proteins through ChIPseq. Comparison of peaks identified in control samples (control shRNA) to those in knockdown samples (*Bcl11a* or *Chd8* shRNA), identified 22198 Bcl11a-specific peaks that were reduced only on *Bcl11a* knockdown, as well as 2149 Chd8-specific peaks that were reduced only on *Chd8* knockdown (Fig. 2D, Appendix Fig. S4A, Dataset EV10). In addition, we found 5089 directly overlapping peaks that were reduced by either *Bcl11a* or *Chd8* knockdown. This subset of shared peaks was in the vicinity of ($+/-1$ kb) 6874 genes.

Strikingly, for this shared peak set, knockdown of either *Bcl11a* or *Chd8* resulted in reduced genomic binding for both proteins, suggesting that Bcl11a and Chd8 are co-dependent for binding at these loci (Fig. 2E; Appendix Fig. S5). Further analysis of all Bcl11a and Chd8 peaks showed that they bind predominantly in enhancer and CpG island regions, with further binding at introns, exons, and promoter regions (Dataset EV11, Appendix Fig. S4C).

Motif enrichment analysis of all Bcl11a and CHD8 peaks also identified strikingly similar GC-rich consensus motifs (Dataset EV12, Appendix Fig. S4D), suggesting that Bcl11a and Chd8 co-bind at

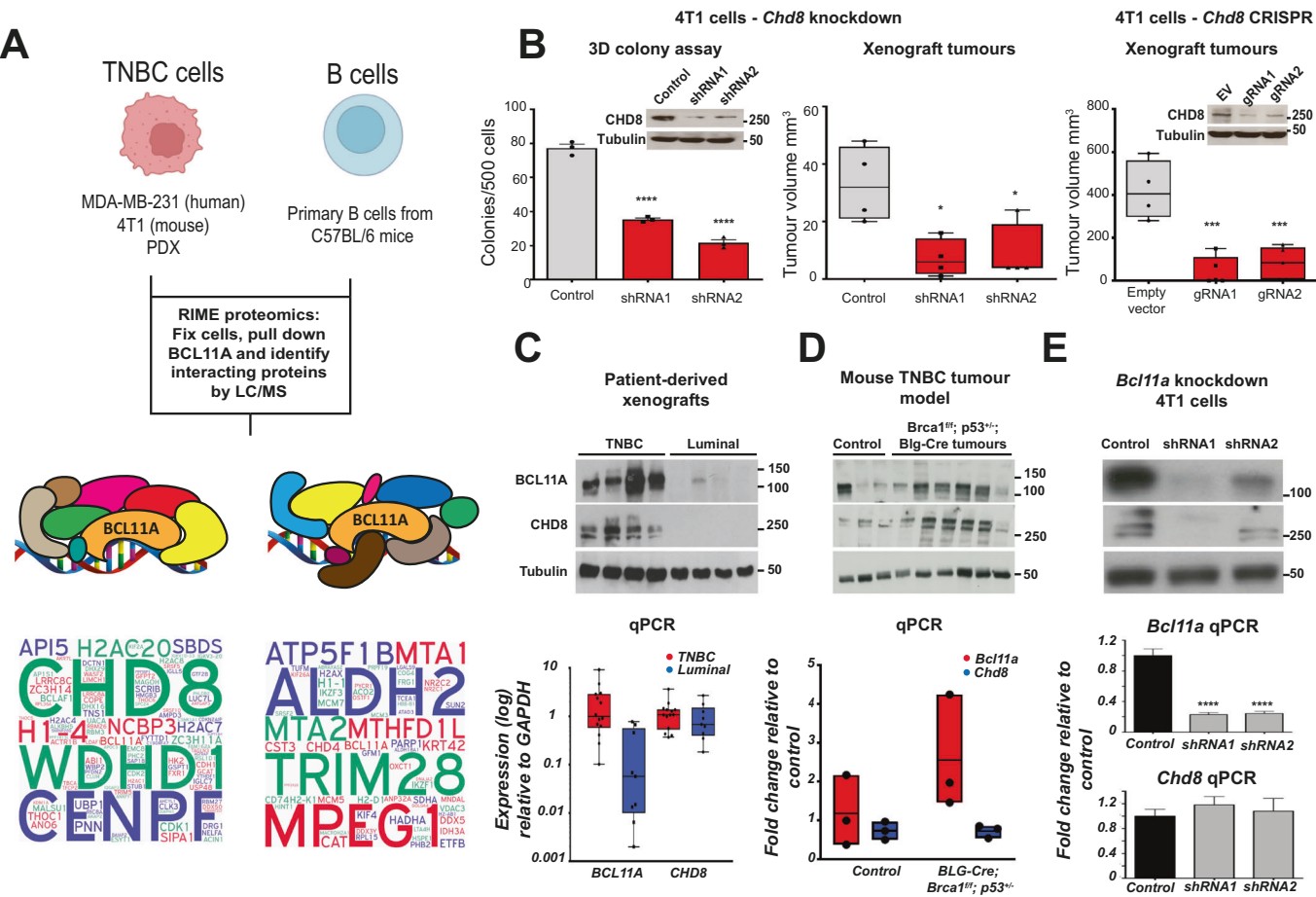

**Figure 1. CHD8 interacts with BCL11A in TNBC and plays a role in TNBC pathology.**

(A) Rapid Immunoprecipitation mass spectrometry of endogenous proteins (RIME) was performed on a range of TNBC samples and primary mouse B-cells, using BCL11A as a bait protein. Detailed results from the mass spectrometry analysis including protein accession number, gene description, mascot score, coverage, and unique peptides have been included in Dataset EV1–6. For cumulative Mascot score word cloud analysis (bottom panels), we first subtracted proteins identified from the BCL11A IP mass spec that were also present in the IgG isotype control IP mass spec, thereby removing any non-specific proteins. Subsequently, proteins with a minimum of one unique peptide were included and the cumulative mascot score was calculated. The word cloud represents each protein and the size representative to the mascot score. This analysis was performed on two independent B-cell runs (right word cloud) and seven independent TNBC cell line or PDX tissue runs (left word cloud). This identified CHD8 as a major interacting partner in TNBC samples but not B-cells, with CHD8 being the only interaction partner identified in all TNBC samples. (B) 4T1 cells were transfected with plasmids encoding either Chd8 shRNA or gRNA sequences and were subsequently used in 3D colony assays or xenografted into mice for tumour growth assays ($n = 3$). An ordinary one-way ANOVA with a Dunnett multiple comparison correction was used. Means of each test condition were compared to the mean of the control condition. P values reported are $*p < 0.05$; $**p < 0.01$; $***p < 0.001$ and $****p < 0.0001$. shRNA-mediated knockdown of Chd8 in mouse TNBC 4T1 cells leads to a reduction in tumour growth in both 3D colony assays (left panel) and xenograft transplantation assays (middle panel), CRISPR-mediated knockout of Chd8 produces similar results (right panel) demonstrating that CHD8 also plays a role in TNBC ($n = 4$). Error bars on the box and whisker denote the minimum and maximum datapoints, the box bounds are the upper and lower quartile values and the line in the middle of the box if is the median value. An ordinary one-way ANOVA with a Dunnett multiple comparison correction was used for statistical analysis of the xenograft plots. Means of each knockdown condition were compared to the mean of the control condition. $*p < 0.05$; $*p < 0.01$; $***p < 0.001$ and $****p < 0.0001$. The Western blot analysis of this figure panel is also shown in Appendix Fig. S1D. (C) Western blot analysis of BCL11A and CHD8 was performed in patient-derived xenografts derived from TNBC or luminal breast tumours. This identified selective upregulation of both proteins in TNBC tumours but not luminal breast cancer tumours. Further qPCR analysis showed amplification of BCL11A RNA in TNBC tumours ($n = 14$) versus luminal tumours ($n = 9$), whereas CHD8 is unaltered at the RNA level. Error bars on the box and whisker denote the minimum and maximum datapoints, the box bounds are the upper and lower quartile values and the line in the middle of the box if is the median value. (D) Western blot analysis of BCL11A and CHD8 was performed using tumours derived from a Brca1f/f p53+/− Blg-Cre mouse model, in which mice spontaneously form TNBC tumours. This showed upregulation of both proteins and upregulation of BCL11A only at the RNA level ($n = 6$). Controls in this experiment were mammary gland preparations from C57BL/6 mice ($n = 3$). Error bars on the box and whisker denote the minimum and maximum datapoints, the box bounds are the upper and lower quartile values and the line in the middle of the box if is the median value. (E) Knockdown of Bcl11a using shRNA in 4T1 cells showed a reduction in the levels of BCL11A and CHD8 protein, with a corresponding reduction in Bcl11a RNA but not Chd8 RNA ($n = 3$). An ordinary one-way ANOVA with a Dunnett multiple comparison correction was used for statistical analysis of these plots. Means of each knockdown condition were compared to the mean of the control condition. $*p < 0.05$; $*p < 0.01$; $***p < 0.001$ and $****p < 0.0001$. The Western Blot in this figure panel is also shown in Appendix Fig. S1D. Source data are available online for this figure.

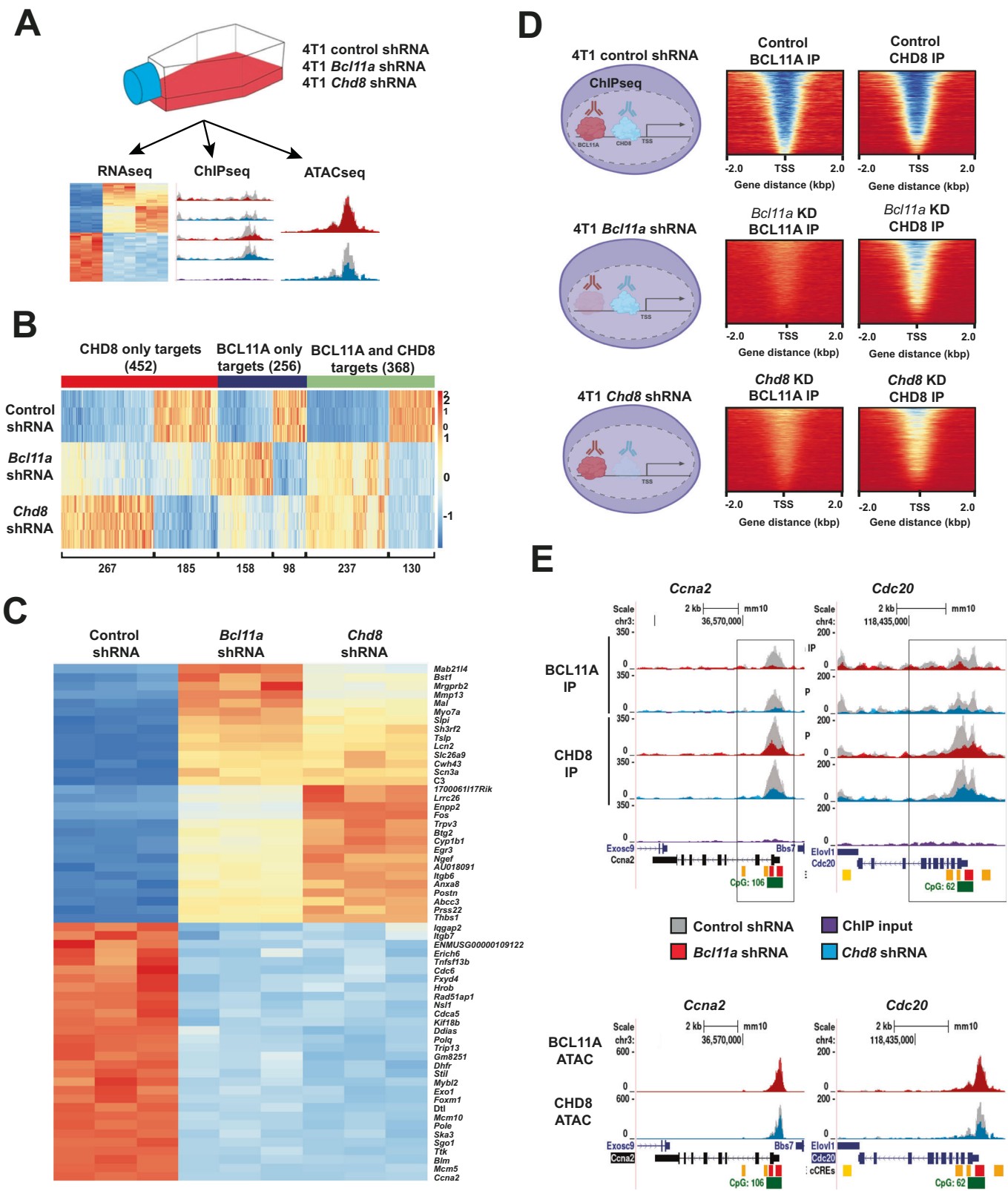

◀ **Figure 2.  Multi-OMICS analysis of Bcl11a and Chd8 knockdown cells identifies a large subset of shared target genes.**

(A) To further explore the role of BCL11A and CHD8 in TNBC, multi-OMICS analysis of 4T1 cells transfected with either control shRNA, Bcl11a-targeting shRNA or Chd8-targeting shRNA was performed. Cells were submitted for bulk RNAseq; ChIPseq, to investigate genomic binding of BCL11A and CHD8; and ATACseq, to investigate modulation of chromatic accessibility. (B) Bulk RNAseq identified a number of protein-specific gene targets, but also identified a large subset of overlapping target genes. The expression of overlapping gene targets appeared to be affected in similar directions and magnitudes by either Bcl11a or Chd8 knockdown. (C) Heat map of the top 30 up- and down-regulated genes from the overlapping gene set. This set of genes is derived from panel B and uses the same scale bar. (D) Transcription start site (TSS) heat maps showing the genomic binding behaviour of BCL11A (left panels) and CHD8 (right panels) in control shRNA (top panels), Bcl11a shRNA (middle panels) and Chd8 shRNA (bottom panels) cell lines. Knockdown of either Bcl11a or Chd8 results in an overall reduction of genomic binding for both proteins around the TSS. (E) Manual inspection of ChIP-seq data confirms co-dependency of BCL11A and CHD8 for binding to promoter regions. Knockdown of either Bcl11a (red peaks) or Chd8 (blue peaks) results in reduced genomic binding versus control (grey peaks). ATACseq analysis shows that knockdown of Chd8, but not Bcl11a, reduces chromatin accessibility at gene target promoter regions.

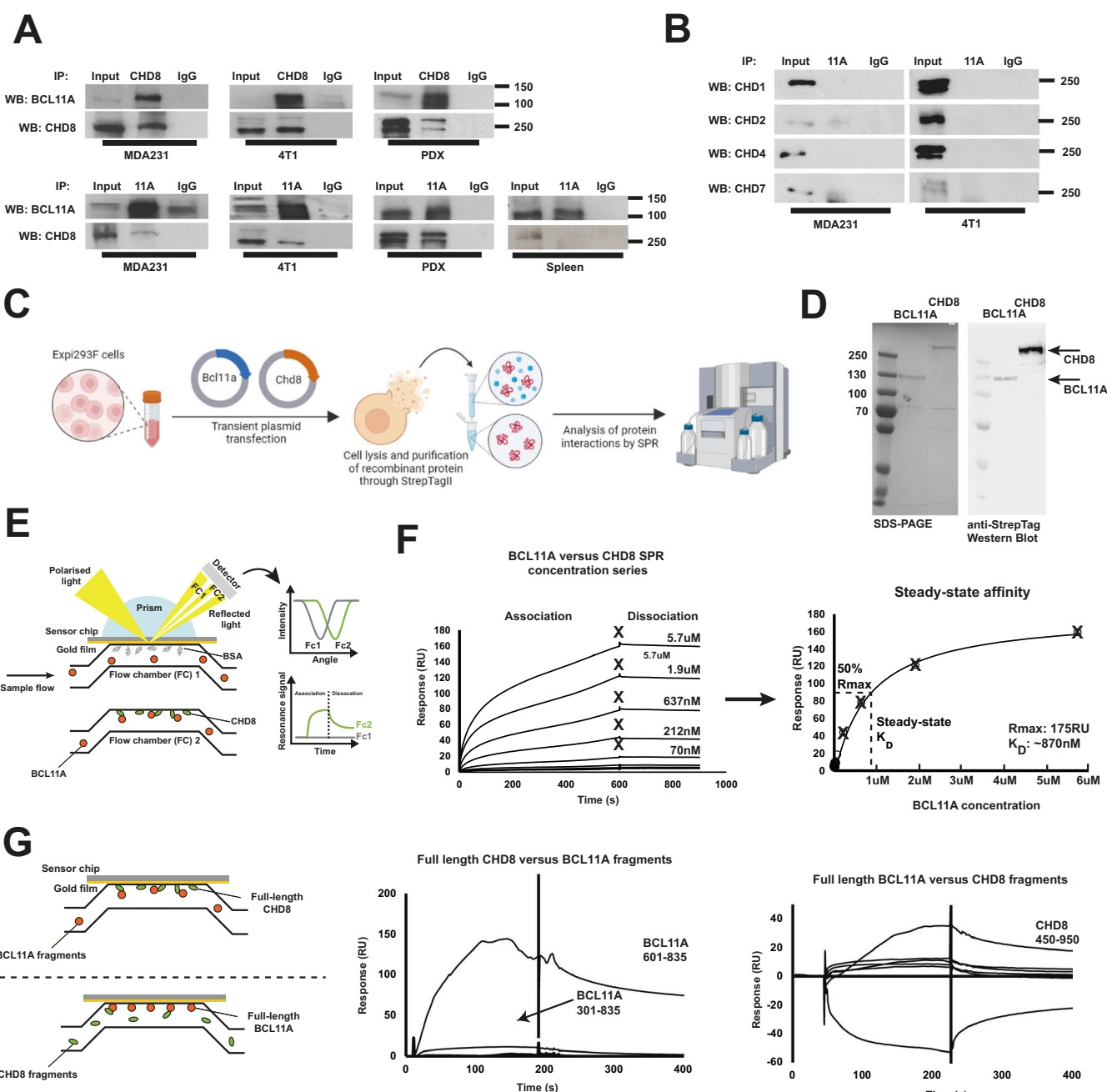

**Figure 3. CHD8 is highly specific for and directly interacts with BCL11A.**

(A) Western blot analysis of BCL11A and CHD8 was performed on samples from co-immunoprecipitation (Co-IP) experiments. BCL11A and CHD8 were independently immunoprecipitated in human TNBC cells (MDA231), mouse TNBC cells (4T1) patient-derived xenografts (PDX) and wild-type (C57BL/6) mouse spleen. Western Blots validated that BCL11A and CHD8 interact in the TNBC and PDX samples, but not in spleen sampes. (B) Western Blot analysis of BCL11A and other members of the CHD family using CoIP samples from panel A. Further analysis showed high specificity of the BCL11A-CHD8 interaction, with no interaction observed between BCL11A and other members of the CHD family. (C) To confirm direct interaction of BCL11A and CHD8, recombinant full-length human versions of BCL11A and CHD8 were expressed and purified from Expi293F cells. Cells were lysed and proteins purified via the StrepTag using StrepTactin XT 4-Flow resin. (D) Purity and quality of recombinant proteins was assessed by SDS-PAGE and Western Blotting, showing a high purity and mono-dispersity of proteins. (E) A surface plasmon resonance (SPR) assay was constructed to confirm direct interaction of BCL11A and CHD8. In this assay, one partner (CHD8) was attached to the surface of an SPR chip, while the other partner (BCL11A) was injected in increasing concentrations. A reference flow cell containing an unrelated protein (BSA) was used as a binding control. Protein binding to the SPR surface causes a change in resonance angle, which can be used to detect protein interactions. (F) SPR using recombinant purified proteins confirms a direct interaction between BCL11A and CHD8. A concentration-dependent increase in binding responses was observed upon injection of BCL11A, with slow association and dissociation rates observed. Responses at the end of the sample injection (600 s) were used to estimate the steady-state affinity of the interaction, which was measured at 870 nM (50% Rmax). (G) A second SPR assay was constructed, whereby either full-length CHD8 or BCL11A was immobilised to the chip surface. Truncated BCL11A or CHD8 proteins were then injected in series into the flow cell containing their interaction partner (i.e. BCL11A vs CHD8 fragments, CHD8 versus BCL11A fragments). This identified BCL11A 601-835 and CHD8 450-950 as the most likely interface between BCL11A and CHD8. The schematic in panel C was generated using BioRender. Source data are available online for this figure.

GC-rich regions. Additional analysis of Bcl11a and Chd8 binding at transcription start sites (TSS) of the shared peak subset confirm that they predominantly bind at or within 4 kb of TSS and likely play a direct role in the transcription of these genes. This analysis also confirmed our earlier observation that knockdown of either *Bcl11a* or *Chd8* globally reduces genomic binding of both proteins at these shared loci (Fig. 2D). To investigate whether genes in the vicinity of these ChIPseq peaks are regulated by Bcl11a and Chd8, we intersected the shared RNAseq gene set (368 genes) with the 6874 genes identified within the vicinity ($+/-1$ kb) of the shared ChIPseq peak set. This identified 78 genes that were represented in both datasets, highlighting genes that are regulated by both Bcl11a and Chd8 at the genomic binding and transcription levels (Appendix Fig. S4B, Dataset EV10). Of these, 38 were downregulated and 40 were upregulated in the absence of either Bcl11a or CHD8.

We then considered our ATACseq dataset (Dataset EV13), which showed that, while knockdown of *Bcl11a* did not affect chromatin accessibility, *Chd8* knockdown influenced accessibility in many shared promoter regions (Fig. 2E; Appendix Fig. S5). In these cases, *Chd8* knockdown resulted in reduced accessibility versus control samples, suggesting that Chd8 partially regulates accessibility of shared promoter regions. The regions affected by CHD8 knockdown in the ATACseq data also overlap with the regions bound by Bcl11a and Chd8 in our ChIPseq dataset, suggesting that Chd8 regulates accessibility at areas bound by both proteins. Taken together, this multiomic dataset reveals that Chd8 and Bcl11a bind promoter regions of their shared target genes, many of which are involved in cell cycle and DNA repair pathways.

## BCL11A and CHD8 directly interact to form a functional complex

We next sought to confirm that BCL11A and CHD8 physically interact to form a functional complex, initially through co-immunoprecipitation (Co-IP) experiments. Co-IP experiments using anti-BCL11A or anti-CHD8 antibodies confirmed that these proteins interact in human (MDA-MB-231) and mouse (4T1) TNBC cell lines as well as patient-derived xenografts, regardless of which protein is used for pulldown (Fig. 3A). To mirror the initial RIME experiments comparing TNBC samples to B-cells, we performed BCL11A Co-IPs using mouse WT spleen samples which are rich in B-cells. This confirmed the expression of BCL11A and CHD8 in this tissue, however, BCL11A was unable to interact with

and pull-down CHD8 in this setting, confirming the specificity of this interaction in TNBC and corroborating the RIME results (Fig. 3A).

Notably, the interaction with BCL11A appears to be specific to CHD8, with additional Co-IP experiments showing that BCL11A does not interact with other CHD family members (Fig. 3B). We explored this further by recombinantly producing and purifying full-length human versions of BCL11A and CHD8 using Expi293F cells (Fig. 3C). Recombinant proteins were tagged and purified using a StrepTagII and StrepTactin XT 4-Flow resin, which resulted in highly pure proteins suitable for use in downstream binding assays (Fig. 3D).

Purified proteins were used in surface plasmon resonance (SPR)-based binding assays (Fig. 3E), in which one interaction partner is immobilised on a sensor surface while the other is injected and allowed to flow freely through the chip. Purified CHD8 was immobilised to the surface of a C1 SPR chip by amine-coupling, followed by the injection of increasing concentrations of purified BCL11A over the chip surface. This confirmed that BCL11A and CHD8 directly interact in a concentration-dependent manner, with an estimated steady-state KD value of 870 nM (Fig. 3F). Notably, the kinetics of this interaction, especially the dissociation rate, were too slow to fit kinetic constants. Visual inspection of the SPR binding data highlights the slow association rate of this interaction, which takes >10 min to reach steady-state, as well as an extremely slow off-rate, suggesting a highly stable complex is formed. To further delineate the BCL11A-CHD8 binding interface, we produced and purified various truncated versions of each protein for use in subsequent SPR binding assays (Appendix Fig. S6A–C).

In these assays, full-length CHD8 was again immobilised to the surface of a C1 SPR sensor chip by amine-coupling with subsequent injection of truncated BCL11A variants (Fig. 3G). Similarly, on a separate sensor surface, full-length BCL11A was immobilised with subsequent injection of truncated CHD8 variants. This identified BCL11A 601-835 as the most likely interacting region with CHD8, and CHD8 450-950 as the most likely interacting region with BCL11A. Interestingly, BCL11A 601-835 comprises a major DNA binding region, whereas CHD8 450-950 contains functional chromatin remodelling domains, suggesting that functional regions of each protein come together to facilitate chromatin remodelling, genomic binding, and subsequent gene transcription (Appendix Fig. S6).

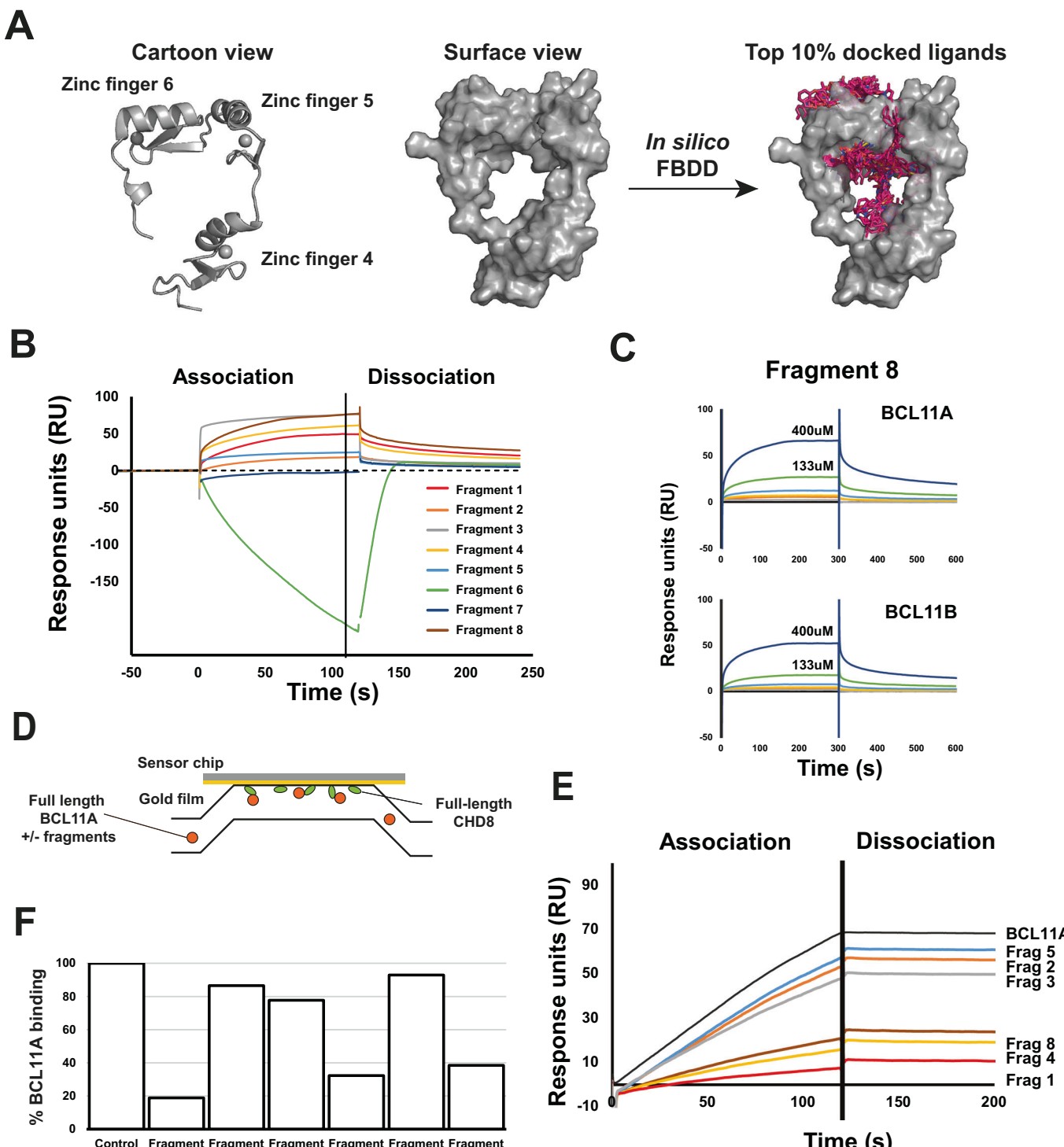

## In silico fragment screening identifies inhibitors of the BCL11A-CHD8 interaction

In an attempt to identify inhibitors of the BCL11A-CHD8 interaction, we performed in silico fragment screening using a previously published crystal structure comprising the three consecutive zinc finger motifs at the C-terminus of BCL11A (residues 731–835, pdb: 6KI6), which is within the vicinity of the CHD8 binding site (Fig. 4A). The top 8 scoring hits from the in silico screen were purchased and validated for BCL11A binding by SPR. This validated 6 of the 8 purchased fragments as BCL11A binders (Fig. 4B). As a counter-screening measure, we then produced and purified the highly homologous family member BCL11B (Appendix Fig. S7) and performed a follow-up concentration series of the 6 validated fragments (Fig. 4C). This showed that most of the BCL11A-binding fragments were more selective for

◄ **Figure 4. In silico fragment screening of BCL11A identifies several hit compounds that prevent in vitro BCL11A-CHD8 complex formation.**

(A) GLIDE-based virtual screening using three diverse fragment libraries was performed using PDB structure 6ki6. The top 10% of hits were re-scored and manually inspected, with the best 8 hits purchased for screening. Hits located in the zinc finger 6 region were prioritised due to higher uniqueness in this region compared to BCL11B. (B) SPR screening for validation of in silico hits. BCL11A was immobilised to an SPR sensor surface, with each fragment injected at a single high concentration (400 µM). 6 out of the 8 purchased fragments were validated as BCL11A binders. (C) Further assessment of validated binders was then performed in a concentration series. Fragments were then injected in a 1:3 serial dilution series to confirm binding and to generate kinetics and affinity data using an SPR chip comprising separate flow cells with either BCL11A or BCL11B immobilised. An example of the concentration series data for fragment 8 is shown for BCL11A and BCL11B binding. All fragments were shown to also bind the homologous BCL11B. (D) An SPR assay was constructed to investigate the ability of fragment hits to prevent BCL11A-CHD8 complex formation. Full-length recombinant CHD8 was immobilised to the chip surface, with subsequent injection of BCL11A alone or of BCL11A that had been pre-incubated with 400 µM of each fragment. Inhibition of the complex was determined through a reduction in binding response of BCL11A binding to the chip surface. (E) All of the fragments tested prevented BCL11A-CHD8 complex formation with varying efficiencies, indicated by a reduction of BCL11A binding to the CHD8-coated surface. (F) Percentage inhibition of BCL11A binding to the chip surface is summarised. Responses are normalised to the BCL11A binding response. Source data are available online for this figure.

BCL11A over BCL11B (Table EV1). We then tested the ability of these fragments to prevent BCL11A-CHD8 complex formation through another SPR assay, in which full-length Chd8 was immobilised to the surface of a C1 SPR sensor chip (Fig. 4D). Full-length BCL11A was then injected either alone or after pre-incubation with each fragment. This showed that all 6 fragments tested affected the ability of BCL11A to interact with CHD8, with the top-performing fragments reducing binding by over 80% (Fig. 4E,F).

## BCL11A-CHD8 interaction inhibitors reduce murine and human TNBC colony sizes in 3D

We then investigated whether these fragments had a functional effect on cells in vitro. To determine this, we treated 4T1 cells for 6 days with the validated fragments and analysed them using 3D colony assays (Fig. 5A,B; Appendix Fig. S9). Average colony sizes were measured and the fold change relative to the average starting sizes on day 0 were calculated per fragment for each timepoint. This identified a significant difference in colony growth rates between the fragments and DMSO control, in particular, fragments 3 and 5 gave the most significant reduction in growth rate compared to DMSO, with an approximate 30% reduction in colony size ($n = 3$, $F = 3.999$, $p = 0.025$) (Fig. 5B; Appendix Fig. S9).

To confirm that this was the result of Bcl11a-Chd8 inhibition, we performed Co-IP experiments using 4T1 cells that had been grown in the presence of fragment 3 or fragment 5 for 24 h (Fig. 5C). As controls, cells were grown in the presence of DMSO only or with a fragment that showed no difference compared to DMSO, fragment 1. An anti-Bcl11a western blot on the 4T1 lysates demonstrated comparable levels of Bcl11a across the different treatments, demonstrating that short-term treatment did not affect Bcl11a expression or turnover. A subsequent Bcl11a IP also showed that the different treatments did not affect the ability of an anti-Bcl11a antibody to recognise Bcl11a. Conversely, when lysates were used in a Chd8 IP, Bcl11a was identified in the DMSO control but not in any of the fragment treatments, suggesting that all fragments identified as inhibitors in the original SPR assay (Fig. 4B) are also able to inhibit this interaction in a cell-based scenario.

To determine if the disruption of Bcl11a-Chd8 affects the proliferation kinetics of the cells, we performed EdU proliferation assays on 4T1 cells treated with the either fragment 1, 3, or 5 for 24 h (Fig. 5D). This analysis showed a marked reduction in the proportion of cells in S phase particularly for the cells treated with fragments 3 or 5, but not fragment 1 (Fig. 5D). This may suggest that the reduction in colony size observed in Fig. 5B can be

attributed to perturbation of the BCL11A–CHD8 interaction, which is also supported by the RNAseq data which suggested that the BCL11A–CHD8 interaction is responsible for driving the expression of multiple cell-cycle and DNA synthesis pathways (Appendix Fig. S3, Dataset EV8 and 9). Based on the initial 4T1 fragment experiments, we then extended 3D colony assay testing into human cell lines using a subset of fragments (Fig. 5A,E; Appendix Fig. S9). To select cell lines for testing, anti-BCL11A and anti-CHD8 Western Blots were performed on a panel of mammary epithelial cells (Appendix Fig. S8). Based on this, the human TNBC cell line MDA-MB-231 was chosen due to strong expression of BCL11A and CHD8, and the luminal breast cancer cell line MCF-7 was selected as a control due to expression of CHD8 but no BCL11A expression. MCF-7 cells showed no difference in colony growth rate ($n = 3$, $F = 0.5817$, $p = 0.6319$) and the intercepts ($n = 3$, $F = 0.5734$, $p = 0.6368$) of the fold changes in colony size over time upon treatment, also seen phenotypically (Fig. 5E). Meanwhile, treatment of the human TNBC cell line (MDA-MB-231) resulted in a difference in colony growth rates across the treatment groups ($n = 3$, $F = 3.727$, $p = 0.0226$) (Fig. 5E), implicating their specific functionality within a TNBC context.

These findings highlight targeting of the BCL11A-CHD8 interaction as a potential new modality for the treatment of TNBC tumours and justifies further development of these fragments into compounds with higher affinity and higher potency.

## Discussion

In this study, we describe a novel TSPPI between the TNBC oncogene BCL11A and the chromatin remodeller CHD8 and showed that like BCL11A, CHD8 also plays a role in TNBC. Using multiomic approaches we have shown that these proteins have a co-dependent relationship in TNBC and that they co-bind and regulate the transcription of several genes associated with cell cycle and DNA repair processes. This reveals an additive oncogenic effect resulting from this novel TSPPI in TNBC. Through in silico fragment-based drug discovery, we identified and validated several chemical entities that bind BCL11A and prevent BCL11A-CHD8 complex formation in vitro.

Our approach in this study coupled the identification of BCL11A interactors through RIME with thorough TSPPI validation and initial drug discovery efforts. The discovery of this interaction opens the possibility of developing a first-in-class targeted TNBC therapy that disrupts the BCL11A-CHD8 complex, for which we have already identified putative lead fragments. However, it is

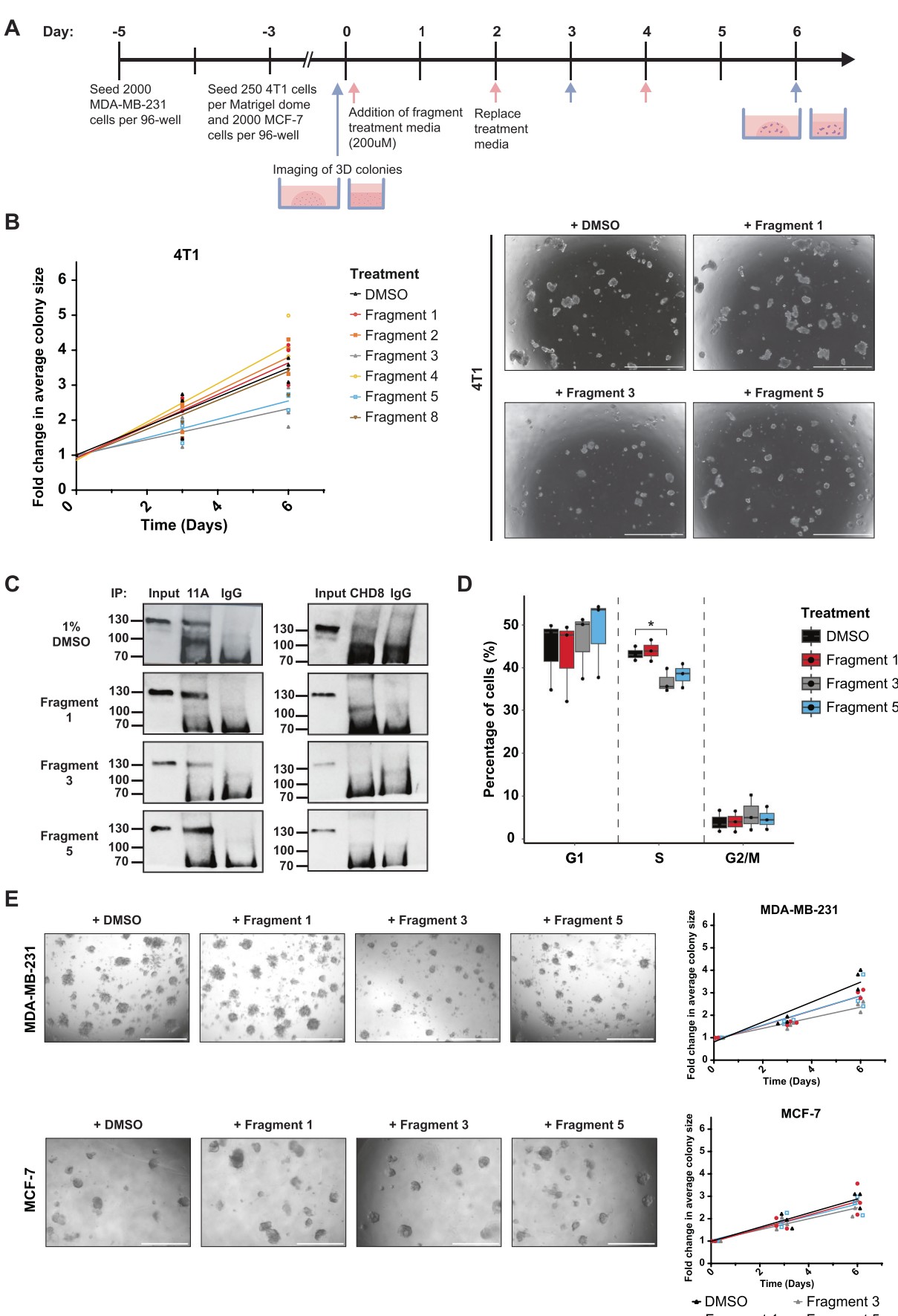

**Figure 5.   3D colony assays identify differential effects on the growth of colony size upon treatment with BCL11A-CHD8 interaction inhibitors, with the effect being TNBC cell-line specific.**

(A) To assess the effect of putative BCL11A-CHD8 interaction inhibitors, 4T1, MCF-7 and MDA-MB-231 cells were grown in 3D and treated with inhibitors across 6 days. Cells were imaged at day 0, 3 and 6 for analysis of colony formation. (B) Fold change in average 4T1 colony size relative to day 0 size upon treatment of fragments at 200 μM across 6 days of treatment, fitted with simple linear regression identifying that the slopes of fold change in average colony size over time differ globally within 4T1s ($n = 3$ passages, F = 3.999, $p = 0.025$) (left panel). 3D colony assay images visualising the phenotypic changes in 4T1 colony size following 6 days of treatment with 3 selected binders. The scale bar represents 2000um (right panel). (C) Co-Immunoprecipitation (CoIP) experiments in which BCL11A and CHD8 were independently immunoprecipitated from 4T1 cells treated with 1% DMSO (vehicle control) or 400 μM of Fragment 1, Fragment 3 or Fragment 5. Anti-BCL11A Western Blots demonstrate that 4T1 treatment has no effect on the levels of BCL11A (input) or in the recognition of BCL11A by an anti-BCL11A antibody (left panels). Conversely, anti-BCL11A Western Blots on CHD8 IP samples (right panels) demonstrates an interaction between CHD8 and BCL11A in the DMSO control, but not in the fragment treatment conditions, suggesting inhibition of the BCL11A-CHD8 interaction by these fragments in a cell-based format. (D) Percentage of 4T1 cells in G1, S and G2/M phase following treatment of fragments 1, 3 and 5 at 200 μM for 24 h, determined by flow cytometry. Data presented as median and range ($n = 3$ passages), G1 analysed by Kruskal–Wallis test, G2/M and S phase analysed by two-way ANOVA with post-hoc Dunnett's test identifying a significant difference between treatment with DMSO and Fragment 3 ($p = 0.0294$). (E) Fold change in average colony size relative to day 0 size comparing human cell lines MCF-7 and MDA-MB-231 upon treatment of fragments 1, 3 and 5 at 200 μM. 3D colony assay images visualising the phenotypic changes in MCF-7 and MDA-MB-231 colony size. The scale bars represent 1000 μm (left panel) Fitted simple linear regressions to fold change in average colony size over time identifies global differences in MDA-MB-231 cells ($n = 3$ passages, F = 3.727, $p = 0.0226$), with no global difference in the slope ($n = 3$, F = 0.5817, $p = 0.6319$) or intercept ($n = 3$ passages, F = 0.5734, $p = 0.6368$) in MCF-7 cells (right panel). Source data are available online for this figure.

important to note that further significant development of these fragments is needed before they can be used as lead drugs in future preclinical studies. Nonetheless, the fragments identified could be used as a basis for the development of new modalities such as PROteolysis TArgeting Chimera (PROTAC) molecules, which do not require inhibitive activity of target-specific molecules and instead simply require handles for further development. The ability to develop these molecules through multiple routes offers flexibility in future efforts to target BCL11A for therapy.

Importantly, the approach described here has broad applicability to other scenarios requiring the identification of PPIs in health or disease. The use of this approach towards targets traditionally considered as "undruggable" is particularly attractive, with the potential to identify new functional partners or PPIs that offer novel routes for therapeutic intervention. Another attractive application of this approach could involve targets that are expressed in multiple tissues naturally or pathologically. In this case, comparison of tissues or pathologies could aid in the identification of protein complexes that exist under certain circumstances. Some of these may also be suitable for therapeutic intervention.

In conclusion, we have reported a proof of principle approach for the identification and targeting of TSPPIs. In the future, we anticipate that this approach will be useful to greatly improve understanding of protein function and for the identification of disease associated PPIs that offer opportunities for therapeutic targeting.

# Methods

### Reagents and tools table

| Reagent/Resource | Reference or Source | Identifier or Catalog Number |
|---|---|---|
| **Experimental models** | | |
| C57BL/6 | The Jackson Laboratory | JAX 000664 |
| Brca1f/f; p53+/-; Blg-Cre | The Jackson Laboratory | JAX 012620 |

| Reagent/Resource | Reference or Source | Identifier or Catalog Number |
|---|---|---|
| MDA-MB-231 | ATCC | HTB-26 |
| 4T1 | ATCC | CRL-2539 |
| Expi293F cells | ThermoFisher Scientific | A14527 |
| MCF-7 | ATCC | HTB-22 |
| **Recombinant DNA** | | |
| piggyBac transposon vector (PB-H1-shRNA-GFP) | System Biosciences | PBSI505A-1 |
| pSF-CMV-Puro-COOH-TEV-Strep | Oxford Genetics | OG1132 |
| pSF-CMV-Puro-NH2-STREP-TEV | Oxford Genetics | OG1131 |
| **Antibodies** | | |
| Anti-BCL11A antibody | Bethyl | A300-382A |
| Anti-rabbit IgG | Cell Signalling Technology | 7074 |
| Anti-BCL11A antibody | Abcam | ab191401 |
| Anti-CHD8 antibody | Novus | NB100-60417 |
| Anti-CHD8 antibody | Abcam | ab220402 |
| Anti-CHD8 antibody | Bethyl | A301-225A-M |
| Anti-CHD1 antibody | Cell Signalling Technology | 4351 |
| Anti-CHD2 antibody | Cell Signalling Technology | 4170 |
| Anti-CHD3 antibody | Cell Signalling Technology | 4241 |
| Anti-CHD4 antibody | Cell Signalling Technology | 11912 |
| Anti-CHD7 antibody | Cell Signalling Technology | 6505 |
| Anti-alpha tubulin antibody | Abcam | ab7291 |
| Anti-StrepTagII antibody | Abcam | ab76949 |
| **Oligonucleotides and other sequence-based reagents** | | |
| Sequences used for shRNA and qPCR primers can be found in the methods (shRNA – shRNA mediated knockdown of BCL11A or CHD8) and Table EV4 (qPCR primers) | | |

| Reagent/Resource | Reference or Source | Identifier or Catalog Number |
|---|---|---|
| **Chemicals, Enzymes and other reagents** | | |
| Dynabeads™ Protein G | Thermo Fisher Scientific | 10004D |
| Transcriptor Reverse Transcriptase | Roche | 3531252103 |
| Reverse transcription system | Promega | A3500 |
| GoTaq Real Time qPCR Master Mix | Promega | A6101 |
| PowerUp SYBR Green Master Mix | Thermo Fisher Scientific | A25742 |
| Qubit RNA HS Assay Kit | Thermo Fisher Scientific | Q32852 |
| NEBNext Ultra II Directional RNA Library Prep Kit | New England Biolabs (NEB) | E7760 |
| ERCC RNA Spike-In Mix | Thermo Fisher Scientific | 4456740 |
| QIAseq FastSelect RNA Removal Kit | Qiagen | 333390 |
| ThruPLEX DNA-seq Kit | Takara Bio | R400674 |
| Nextera DNA library preparation kit | Illumina | FC-121-1031 |
| StrepTactin XT 4-Flow resin | IBA Biosciences | 2-5010-025 |
| 2 ml Pierce centrifuge columns | Thermo Fisher Scientific | 89896 |
| Vivaspin 20 30 K MWCO Centrifugal Concentrator | Sartorius | 10506282 |
| Slide-A-Lyzer 0.5 ml 10 K MWCO dialysis cassette | Thermo Fisher Scientific | 66383 |
| Sensor Chip C1, Series S | Cytiva (previously GE) | 29104944 |
| Sensor Chip CM5, Series S | Cytiva (previously GE) | 29104988 |
| **Software** | | |
| Proteome Discoverer v1.3 | Thermo Scientific | |
| Mascot search engine v2.3.0 | Matrix Science | |
| Biacore T200 Evaluation Software | Cytiva | |
| FastQC v0.11.8 | Babraham Bioinformatics | |
| multiQC v1.9 | Seqera | |
| STAR v2.7.0a | Dobin et al, 2013 | |
| featureCounts | Liao et al, 2014 | |
| noisyR | Moutsopoulos et al, 2021 | |
| bulkAnalyseR | Moutsopoulos et al, 2023 | |
| edgeR | Robinson et al, 2010 | |
| DESeq2 | Love et al, 2014 | |
| gProfiler2 | Raudvere et al, 2019 | |
| bowtie2 v2.4.2 | Langmead and Salzberg, 2012 | |

| Reagent/Resource | Reference or Source | Identifier or Catalog Number |
|---|---|---|
| macs2 v2.2.7.1 | Zhang et al, 2008 | |
| pyBigWig v0.3.17 | Ramirez et al, 2016 | |
| GOLD suite v5.3 | CCDC, Cambridge | |
| ImageJ v1.54f | NIH | |
| FlowJo v10.10.0 | FlowJo | |
| **Other** | | |
| New Brunswick™ s41i CO$_2$ Incubator Shaker | Eppendorf | |
| Bioruptor® Pico | Diagenode | |
| LTQ Velos-Orbitrap MS | Thermo Scientific | |
| Ultimate RSLCnano-LC system 34 | Dionex | |
| Step-One Plus Real-Time PCR system | Thermo Fisher Scientific | |
| TapeStation D5000 | Agilent Technologies | |
| HiSeq4000 | Illumina | |
| Biacore T200 | Cytiva | |
| Cell Sorter SH800 | Sony | |

## Cell line maintenance

EpH4 (Merck, SCC284), 4T1 (ATCC, CRL-2539), MDA-MB-231 (ATCC, HTB-26) and MCF-7 (gifted from the Carroll Lab, Cancer Research UK Cambride Institute (CRUK CI)), cell lines were grown in DMEM-F12 (Gibco, 21331-020), RPMI (Gibco, 52400-025) and DMEM (Gibco, 41965-039), respectively, supplemented with 10% foetal bovine serum, 1% Pen Strep (Gibco), and maintained at 37 °C and 5% CO$_2$. Cells were passaged at near confluency using split ratios of 1:4, 1:10 and 1:6, respectively. Expi293F (Thermo, A14527) cells were used for recombinant protein expression and were cultured in Expi293F expression media (Thermo, A1435101) at 37 °C, 8% CO$_2$ and 125 rpm. Expi293F were cultured to densities of $3-5 \times 10^6$ and were then counted, assessed for viability, and subcultured to densities of $3-5 \times 10^5$.

## Rapid immunoprecipitation and mass spectrometry of endogenous proteins (RIME)

The RIME experiment using cells and tissue was conducted as published previously (Mohammed et al, 2016, 2013). Briefly, TNBC cells ($1 \times 10^7$) were grown in their respective complete media. The media was then removed and replaced with media containing 1% formaldehyde (Fisher Scientific, 28906) and cross-linked for 8 min. Cross-linking was quenched by replacing the formaldehyde with 0.2 M Glycine in PBS. The cells were washed twice with ice-cold PBS and harvested in PBS containing 1x protease inhibitors (Roche). For processing PDX tissue, the sample was first chopped into fine pieces and then cross-linked for 20 min in 20 ml of solution A (1% formaldehyde, 50 mM Hepes-KOH, 100 mM NaCl, 1 mM EDTA, 0.5 mM EGTA in PBS). Cross-linking was quenched by adding 1/20 volume of 2.5 M Glycine. The tissue was then centrifuged at $2000 \times g$ for 5 min and resuspended in cold PBS.

The sample was then homogenised using a douncer and resuspended in cold PBS. The resulting pellet after another centrifugation was then used for steps below.

Primary B-cells derived from C57BL/6 mice were processed in an identical manner and were prepared as previously described (Yu et al, 2012). Briefly, cells from bone marrow (femoral bone), spleen, thymus and lymph node were mechanically dissociated, and the red blood cells removed using ACK lysis buffer (Lonza). B-cells were sorted from the population as previously described and were cultured in MethoCult M3630 (Stem Cell Technologies), according to the manufacturer's instructions, at 37 °C with 5% $CO_2$. As described above, $1 \times 10^7$ cells were then used for RIME.

The nuclear fraction was extracted by first resuspending the pellet in 10 ml of LB1 buffer (50 mM HEPES-KOH [pH 7.5], 140 mM NaCl, 1 mM EDTA, 10% glycerol, 0.5% NP-40 or Igepal CA-630, and 0.25% Triton X-100) for 10 min at 4 °C. Cells were then pelleted by centrifugation at $2000 \times g$ for 5 min at 4 °C. The supernatant was discarded, and the pellet was resuspended in 10 ml of LB2 buffer (10 mM Tris-HCL [pH 8.0], 200 mM NaCl, 1 mM EDTA, and 0.5 mM EGTA) and mixed at 4 °C for 5 min. Cells were again centrifuged at $2000 \times g$ for 5 min at 4 °C. The supernatant was discarded, and the pellet resuspended in 300 μl of LB3 buffer (10 mM Tris-HCl [pH 8], 100 mM NaCl, 1 mM EDTA, 0.5 mM EGTA, 0.1% Na-deoxycholate, and 0.5% N-lauroylsarcosine) and sonicated in a waterbath sonicator at 4 °C using a 30 s on/off cycle for 10 min (Diagenode Pico Bioruptor). After sonication, 30 μl of 10% Triton X-100 was added and the lysate clarified by centrifugation at $20,000 \times g$ for 10 min at 4 °C. The supernatant was kept and used as the input for the RIME.

To set up magnetic beads for IP, 100 μl bead suspension was added into each required Eppendorf tube and the tubes were placed into a magnetic rack. The storage buffer was removed, and the beads resuspended in 1 ml PBS supplemented with 5 mg/ml BSA to wash and block the beads. Tubes were again placed on the rack and the supernatant removed. This process was repeated three times for a total of four washes. Beads were then suspended in 500 μl PBS/BSA + 10 μg relevant antibody and were mixed gently overnight on a rotator set to 20 rpm at 4 °C. The BCL11A antibody (A300-382A, Bethyl) was used for the experiments and as controls, an IgG antibody (anti-rabbit IgG 7074, CST).

The supernatant generated from cell lysis was then incubated with the 100 μl of magnetic beads (PureProteome Protein G, Sigma LSKMAGG02) prebound with 10 μg antibody, and immunoprecipitation (IP) was conducted overnight at 4 °C. The beads were washed ten times in 1 ml of RIPA buffer and twice in 100 mM ammonium hydrogen carbonate (AMBIC) solution. After the second AMBIC wash, the beads were transferred to new tubes and AMBIC removed. The beads were stored at −20 °C until ready for tryptic digestion.

Tryptic digestion of bead-bound protein was performed by adding 10 μl of trypsin (10 μg/ml in 100 mM $NH_4HCO_3$) directly onto the washed beads. After a brief vortex to resuspend the beads, they were incubated at 37 °C overnight. The tubes were placed on a magnet to separate the beads from the tryptic digest and the supernatant was removed. Supernatant was then placed into formic acid to give a final concentration of 5%. Digested peptide mixtures were diluted typically 1:10 with loading buffer (0.1% formic acid/2% acetonitrile/water), and 2–5 μl aliquots of the diluted peptide mixture was analysed by nano-LC-MS/MS.

Mass spectrometry (MS) was performed using an LTQ Velos-Orbitrap MS (Thermo Scientific) coupled to an Ultimate RSLCnano-LC system 34 (Dionex). Optimal separation conditions resulting in maximal peptide coverage were achieved using an Acclaim PepMap 100 column (C18, 3 μm, 100 Å) (Dionex) with an internal diameter of 75 μm and capillary length of 25 cm. A flow rate of 350 nl/min was used with a solvent gradient of 5% B to 50% B in 120 min. Solvent A was 0.1% (v/v) formic acid in water, whereas the composition of solvent B was 80% (v/v) acetonitrile in 0.1% (v/v) formic acid. The mass spectrometer was operated in positive ion mode using an $N^{th}$ order double play method to automatically switch between Orbitrap-MS and LTQ Velos-MS/MS acquisition. Survey full-scan MS spectra (from 400 to 1800 $m/z$) were acquired in the Orbitrap with resolution (R) 60,000 at 400 $m/z$ (after accumulation to a target of 1,000,000 charges in the LTQ). The method used allowed sequential isolation of the 20 most intense ions for fragmentation in the linear ion trap, depending on signal intensity, using CID at a target value of 5000 charges. For accurate mass measurements, the lock mass option was enabled in MS mode, and the 445.120025 ion was used for internal recalibration during the analysis. Target ions already selected for MS/MS were dynamically excluded for 60 s. General MS conditions were electrospray voltage, 1.56 kV with no sheath or auxiliary gas flow, an ion selection threshold of 1000 counts for MS/MS, an activation Q value of 0.25, activation time of 10 ms, capillary temperature of 250 °C, and an S-Lens RF level of 60% were also applied. Charge state screening was enabled, and precursors with unknown charge state or a charge state of 1 were excluded.

Raw MS data files were processed using Proteome Discoverer v.1.3 (Thermo Scientific). Processed files were searched against the SwissProt human database 35 using the Mascot search engine version 2.3.0. Searches were done with tryptic specificity allowing up to one miscleavage and a tolerance on mass measurement of 10 ppm in MS mode and 0.6 Da for MS/MS ions. Structure modifications allowed were oxidized methionine, and deamidation of asparagine and glutamine residues, which were searched as variable modifications. Using a reversed decoy database, false discovery rate (FDR) was less than 1%.

Detailed results including protein accession number, gene description, mascot score, coverage, and unique peptides have been included in Dataset EV1–6. For cumulative Mascot score word cloud analysis, we first subtracted proteins from the BCL11A IP list that were also present in the IgG IP list, thereby removing any non-specific proteins. Subsequently, proteins with a minimum of one unique peptide were included and the cumulative mascot score was calculated. The word cloud represents each protein and the size representative to the mascot score. This analysis was performed on two independent B-cell runs and seven independent TNBC cell line or PDX tissue runs.

## Western blotting

Cells were lysed using RIPA (Cell Signalling) and 1x protease inhibitors (Roche) as per manufacturer instructions. For tissue lysis, mammary or tumour tissue was pulverised using a mortar and pestle in liquid nitrogen. The pulverised sample was dissolved in RIPA after which the sample was passed through a 25 G needle.

Total protein was measured using the bicinchoninic acid (BCA) method (Pierce Biotechnology). In total, 50 μg cell lysate was separated using 7.5% SDS-PAGE gels and transferred to PVDF membranes (Amersham) by electro-blotting. Membranes were blocked in 5% (w/v) skimmed milk powder in Tris-buffered saline (20 mM Tris, 150 mM NaCl, pH 7.6) containing 0.1% Tween-20 (TBST). Blots were then incubated at 4 °C overnight with primary antibodies as indicated, washed 3x for 5 min in TBST and were subsequently probed with secondary antibodies for 1 h at room temperature. Membranes were then washed again 5x for 5 min in TBST. After removal of wash buffer, ECL solution (Amersham) was then added to the membrane and chemiluminescence analysed. Antibodies used were, anti-BCL11A (ab191401, Abcam, 1:3000), anti-CHD8 (NB100-60417, Novus, 1:10,000; or ab220402, abcam 1:1000), anti-CHD1 (4351, CST, 1:1000), anti-CHD2 (4170, CST, 1:1000), anti-CHD3 (4241, CST, 1:1000), anti-CHD4 (11912, CST, 1:1000), CHD7 (6505, CST, 1:1000) and anti-alpha-Tubulin (ab7291, Abcam, 1:10,000). Side-by-side comparisons of the cropped Western Blot figures presented in the main paper figures alongside the original X-ray films they are derived from can be found in Source data files.

## Co-immunoprecipitation

Cells were lysed using RIPA (Cell Signalling) and 1x protease inhibitors (Roche) as per the manufacturer's instructions. Total protein was measured using the BCA method as described above (Pierce Biotechnology). Briefly, 500 μg cell lysates were added to 50 μl Dynabeads Protein G (Thermo Fisher Scientific) and were pre-cleared for 3 h at 4 °C to remove non-specific binding. Pre-cleared lysates were then incubated with anti-BCL11A (Bethyl, A300-382A), anti-CHD8 (NB100-60417, Novus) or control IgG (anti-rabbit IgG 7074, CST) at 4 °C overnight. The following day, 50 μl of Dynabeads Protein G were added to each sample. After 3 h, the beads were washed three times with RIPA buffer, and were analysed by Western Blot as described above. For fragment CoIP experiments, the protocol above was followed after cells had been treated with 400 μM of each fragment in a final concentration of 1% DMSO for 24 h. A 1% DMSO condition was used as a vehicle control.

## Mouse experiments

Brca1f/f; p53+/-; Blg-Cre (JAX 012620) mice were used for the study of TNBC tumours. Animals were monitored for tumour growth and tumour volume was measured every 48–72 h using calipers. All tumours were collected, processed, and stored in an identical manner. Briefly, tumours that exceeded 1.2 cm² were excised from humanely killed mice and were snap frozen on dry ice for isolation of protein and RNA. Tumours were only observed in the mammary epithelium in the Brca1f/f; p53+/-;Blg-Cre mice.

The oestrus cycle stage of nulliparous animals was determined by vaginal smears. All mice were housed in individually ventilated cages under a 12:12 h light–dark cycle, with water and food available *ad libitum* and were euthanized by terminal anaesthesia. All experimental animal work was performed in accordance to the Animals (Scientific Procedures) Act 1986, UK and approved by the Ethics Committee at the Sanger Institute.

## Patient-derived xenografts (PDX)

Material from the previously described BioBank of PDXs was used in this study (Bruna et al, 2016) (see Table EV2 and EV3). Protein and RNA was provided by the BioBank and processed for western blot or qPCR as above.

## Preparation of RNA

RNA was extracted using the RNeasy mini kit (Qiagen), according to the manufacturer's instructions. For cell lines, T25 flasks were first washed with cold PBS, and 350 μl of buffer RLT was added. Cells were then scraped and passed through a 20 G syringe five times. For RNA preparation from sorted cells, cells were spun down and resuspended in buffer RLT, and RNA was extracted using the RNeasy mini kit (Qiagen) according to the manufacturer's instructions. For tissue lysis, mammary or tumour tissue was pulverised using a mortar and pestle under liquid nitrogen. The pulverised sample was dissolved in Trizol (Thermo Fisher Scientific) and purified using the RNeasy mini kit (Qiagen). For all preparations, DNA was degraded by adding 20U RNase-free DNaseI (Roche) for 30 min at room temperature. DNaseI treatment was performed on column. RNA was quantified by nanodrop (Thermo).

## Preparation of cDNA and qRT-PCR

1 μg of total RNA was diluted to a final volume of 11 μl in RNase-free H₂O. 2 μl of random primers (Promega) were then added after which the mixture was incubated at 65 °C for 5 min. A master mix containing Transcriptor Reverse Transcriptase (Roche), Reverse Transcriptase buffer, 2 mM dNTP mix and RNasin Ribonuclease Inhibitors (Promega) was added to each sample. This mixture was incubated at 25 °C for 10 min, then 42 °C for 40 min and finally 70 °C for 10 min. The resulting cDNA was then diluted 1:2.5 in H₂O for subsequent use. qPCR was performed using a Step-One Plus Real-Time PCR System (Thermofisher Scientific). Either Taqman (ThermoFisher Scientific) probes with GoTaq Real Time qPCR Master Mix (Promega) or primers (Sigma) with PowerUp SYBR Green Master Mix (ThermoFisher Scientific) were used. Enrichment was normalised with control mRNA levels of *GAPDH* and relative mRNA levels were calculated using the ΔΔCt method comparing to control group. For the list of primers and probes see Table EV4. For statistical analysis of mRNA expression, an ordinary one-way ANOVA with a Dunnett multiple comparison correction was used. Means of each test condition were compared to the mean of the control condition. $P$ values reported are $*p < 0.05$; $**p < 0.01$; $***p < 0.001$ and $****p < 0.0001$.

## shrna-mediated knockdown of BCL11A or CHD8

*BCL11A* shRNA sequences (TRCN0000033449 and TRCN0000033453), h*CHD8* (TRCN0000367896 and TRCN0000360109) and m*Chd8* (TRCN0000217316 and TRCN0000241054) were obtained from TRC consortium and cloned into a piggyBac transposon vector (PB-H1-shRNA-GFP) as described previously (Khaled et al, 2015). MDA-MB-231 and 4T1 cells grown in 25 cm² tissue culture flasks were transfected with 4 μg of respective vector using Lipofectamine 3000 or Lipofectamine LTX (Invitrogen). Cells were treated with G418 (400 μg/mL) (Gibco) for

5 days after which GFP$^+$ cells were sorted and cultured. Knockdown was confirmed by Western Blotting and qPCR as described above.

## 3D colony assays

To investigate the effect of *Chd8* knockdown on 4T1 colony formation, 500 control 4T1 cells or 500 CHD8 knockdown 4T1 cells were resuspended in matrigel (BD Biosciences) and were seeded into a 6-well plate. The plate was then incubated for 15 min at 37 °C to allow hardening of suspension. Growth media was added to the well and changed every 2–3 days for 20 days. All experiments were performed in triplicates. For statistical analysis of colony number per 500 cells seeded, an ordinary one-way ANOVA with a Dunnett multiple comparison correction was used. Means of each test condition were compared to the mean of the control condition. $P$ values reported are *$p < 0.05$; **$p < 0.01$; ***$p < 0.001$ and ****$p < 0.0001$.

## Xenograft transplantation assays

500,000 4T1 cells were suspended in 25% matrigel (BD Biosciences) and HBSS. This mixture was subcutaneously injected in 6–12-week-old NSG mice. Animals were monitored for tumour growth and tumour volume was measured every 48–72 h using calipers. Cells were allowed to grow in the animals for a maximum 9 weeks (63 days) post injection or until the tumour reached a maximum allowed area of 1.2 cm$^2$. Importantly the experiment was double blind—cells were assigned random names by one experimenter who handed them over to a second experimenter who then injected the animals and tumours were measured by an animal technician who was unaware of the identity of the lines. For statistical analysis of xenograft tumour sizes, an ordinary one-way ANOVA with a Dunnett multiple comparison correction was used. Means of each test condition were compared to the mean of the control condition. $P$ values reported are *$p < 0.05$; **$p < 0.01$; ***$p < 0.001$ and ****$p < 0.0001$.

## Generation of multiOMIC dataset

### RNAseq

RNA was submitted for library preparation by the Wellcome-MRC Cambridge Stem Cell Institute. Initial QC was performed using Qubit RNA HS Assay Kit (Thermo Fisher Scientific) and TapeStation Analysis software (Agilent Technologies). 1000 ng RNA was used for library preparation using the NEBNext Ultra II Directional RNA Library Prep Kit (NEB), ERCC spike-ins, and the QIAseq FastSelect RNA Removal Kit (Qiagen). Libraries were subsequently measured using the Qubit system and visualized on a TapeStation D5000. Sequencing was performed as one lane of PE150 using the HiSeq4000 at the CRUK-CI Genomics Core.

### ChIPseq

ChIP-Seq experiments were performed as previously described (Schmidt et al, 2009). Antibodies used were BCL11A (Bethyl, A300-382A and Abcam ab191401) and CHD8 (Novus NB100-60417 and Bethyl A301-225A-M). Briefly, 2 × 15 cm plates per cell line were formaldehyde cross-linked, the nuclear fraction isolated and chromatin sonicated using Bioruptor Pico (Diagenode). IP was performed using 100 μl of Dynabeads Protein G (Thermo Fisher

Scientific) and 10 μg of antibody. The samples were then reverse cross-linked and DNA was eluted using Qiagen MinElute column. Samples were then processed for library prep and sequencing. ChIP-Seq libraries were prepared using the ThruPLEX DNA-seq Kit. The pool was run as a PE150 run type on one lane of a HiSeq4000.

### ATACseq

ATACseq assay were performed using Illumina kits. Briefly, a homogenous cell pellet was lysed and resuspended in a transposition mix. The samples were then incubated at 37 °C for 30 min and then purified using the Qiagen MinElute PCR Purification Kit. Eluted sample were then processed for library prep and sequencing. Libraries were prepared using the PCR steps (12 cycles) in the Illumina Nextera DNA library preparation kit. The pool was run as a PE150 run type on one lane of a HiSeq4000.

## Production and purification of recombinant proteins

For this study, full-length human versions of CHD8, BCL11A and BCL11B, as well as truncated versions of CHD8 and BCL11A, were recombinantly produced and purified for use in binding assays. Genes encoding for full-length *CHD8, BCL11A* and *BCL11B* were first sub-cloned from existing plasmids into the pSF-CMV-Puro-COOH-TEV-Strep plasmid (Oxford Genetics, OG1132) by InFusion cloning (Clontech), following the manufacturer's instructions. Truncated variants were instead cloned into the OG1132 vector by restriction digest and ligation cloning using T4 DNA ligase (NEB).

Sequence-verified plasmids were then transiently transfected into Expi293F cells (Thermo Fisher Scientific) using transfection grade linear polyethylenimine hydrochloride (MW 40000) (PEI MAX, Fisher Scientific, NC1038561). To form PEI:DNA complexes, 37.5 μg plasmid and 150 μg PEI MAX were separately diluted to 1.25 ml in Expi293F expression media. Diluted plasmid and PEI solutions were then mixed and complexes allowed to form for 30 min at room temperature. Complexes were then added dropwise to 22.5 ml of 3–5 × 10$^6$ Expi293F cells. The following day, an additional 25 ml of Expi293 expression medium was added. Cells were allowed to express protein for a total of 48 h and were subsequently collected by centrifugation at $500 \times g$ for 5 min. Supernatant was removed and cell pellets were washed 3x with 25 ml sterile ice-cold PBS by centrifugation at $500 \times g$ for 5 min. After the final wash, pellets were snap frozen and stored at −80 °C until use.

For purification, each 50 ml cell pellet derived from the above protocol was lysed in 10 ml lysis buffer (100 mM Tris, 0.5 M NaCl, 1% NP-40 [IGEPAL CA-630], 1% sodium deoxycholate, 0.5 M Arginine hydrochloride, 25 U/ml Benzonase nuclease, 1x cOmplete protease inhibitor, pH adjusted to 8 using concentrated HCl). Cells were first resuspended in lysis buffer by vortexing and were then drawn through a blunted 21 g needle 8x. Lysate was incubated on ice for 45 min and was then clarified by centrifugation at $15,000 \times g$ for 15 min at 4 °C. Supernatant was decanted into a clean tube and was centrifuged again at $15,000 \times g$ for 15 min at $4 \times g$. Supernatant was then decanted into a clean tube and stored on ice until ready for use.

StrepTactin XT 4-Flow resin (IBA Biosciences, 2-5010-025) was used for protein purification. For each lysed 50 ml cell pellet, 5x 1 ml gravity flow columns containing StrepTactin XT 4-Flow resin

were constructed using 2 ml Pierce centrifuge columns (Thermo, 89896). Columns were constructed and stored at 4 °C. Columns were used in gravity flow format such that buffer was added gently to the top of the resin bed, taking care not to disturb the resin, and columns were allowed to empty by gravity flow. Each column was first equilibrated with 3x 2 ml washes using wash buffer (100 mM Tris, 0.5 M NaCl, 0.5 M Arginine hydrochloride, 0.05% Tween 20, pH 8). The 10 ml lysate was then split equally over 5 columns (i.e. 2 ml/column) and allowed to empty by gravity flow. The lysate was reapplied once more and similarly allowed to empty by gravity flow. Columns were then washed with consecutive rounds of 2 ml wash buffer until the absorbance of the wash at 280 nm was consistently <0.01, as measured by nanodrop (Thermo). Protein was then eluted in 6 fractions of 0.5 ml with elution buffer (100 mM Tris, 0.5 M NaCl, 0.5 M Arginine hydrochloride, 0.05% Tween 20, 50 mM Biotin, pH 8). Columns were regenerated with 5x 2 ml washes of freshly prepared 10 mM NaOH and were then requilibrated with 3x 2 ml washes using wash buffer. Columns were then subsequently stored in wash buffer until next use.

All fractions from the 5 columns were combined (15 ml total) and were concentrated using a Vivaspin 20 30 K MWCO centrifugal concentrator (Sartorius) at $2000 \times g$ and 4 °C. The sample was mixed by pipetting every 10 min and samples were concentrated until they were <0.5 ml in volume. Sample was then recovered from the concentrator and added into a Slide-A-Lyzer 0.5 ml 10 K MWCO dialysis cassette (Thermo, 66383) that had been pre-wetted following the manufacturer's instructions in HBS-P+ buffer (10 mM HEPES, 150 mM NaCl, 0.05% Tween 20, pH 7.4). Proteins were then dialysed against HBS-P+ overnight with gentle stirring at 4 °C, with a single complete buffer change after 2 h. Volume of dialysis buffer was 500x that of the sample. The following day samples were recovered from the dialysis cassette and their absorbance at 280 nm measured versus dialysis buffer to estimate protein concentration. Proteins were then aliquoted, snap frozen in LN2 and stored at –80 °C until use. Data used to generate SPR figures can be found in Dataset EV15–21 and Table EV6.

## Surface plasmon resonance (SPR) analysis of the BCL11A-CHD8 interaction

All SPR experiments were performed using a Biacore T200 instrument (Cytiva). For analysis of the BCL11A-CHD8 interaction, a new sensor chip C1 (Cytiva, 29104944) was docked into the instrument and the system primed with HBS-P+ buffer (10 mM HEPES, 150 mM NaCl, 0.05% Tween 20, pH 7.4). The chip was then conditioned with 2× 1-min injections of conditioning solution (0.1 M Glycine-NaOH, 0.3% Triton X-100, pH 12), and the system was again primed in HBS-P+.

Immobilisation of CHD8 to the chip surface was performed in a manual run, using parameters flowpath 1,2; flow rate 10 µl/min and no reference subtraction. The surface was then activated with a 420 s injection of a N-hydroxysuccinimide(NHS)/1-ethyl-3 (3-dimethylaminopropyl) carbodiimide hydrochloride (EDC) mixture (NHS/EDC), in which stock solutions of 11.5 mg/ml NHS and 75 mg/ml EDC were mixed in equal volumes immediately before use. After activation, the flow path was changed to 1 and a 50 µg/ml dilution of BSA resuspended in 10 mM sodium acetate pH 4 was injected for 20 min. The flow path was then changed to 2 and a 1/10 dilution of 0.35 mg/ml CHD8 in 10 mM sodium acetate pH 4 was injected for

20 min. The flow path was then changed to 1,2 and the surface inactivated with a 420 s of 1 M Ethanolamine hydrochloride-NaOH pH 8.5, giving final immobilisation levels of 760RU and 1585RU, respectively.

Interaction analysis was performed using multi-cycle kinetics (MCK). Data was collected at a rate of 10 Hz using multi detection, with a sample compartment temperature of 4 °C, an analysis temperature of 25 °C, a flow rate of 30 µl/min and HBS-P+ as the running buffer. 10 startup cycles were used to allow for baseline stabilisation, which comprised 10 cycles of a 600 s injection of buffer followed by a 300 s dissociation time and a 60 s injection of regeneration buffer (10 mM NaOH). BCL11A was then injected in a 1:3 serial dilution series of increasing concentration, ranging from 2.62 nM to 5737.5 nM, and included a zero-concentration injection (0 nM, buffer only). Sample cycles consisted of a 600 s association phase, a 300 s dissociation phase, and a 60 s regeneration phase. The BSA-coupled flow cell was used as a reference, and zero-concentration controls were used for double referencing. All analysis was carried out using BIAcore T200 evaluation software v1.0. Steady-state affinity models were then fitted to reference subtracted sensorgrams to calculate steady-state $K_D$ values. Data used to generate SPR figures can be found in Dataset EV15–21 and Table EV6.

## SPR analysis of truncated BCL11A and CHD8 variants

For analysis of the truncated BCL11A and CHD8 variants, two series S CM5 sensor chips were constructed—one for testing of truncated CHD8 variants, and one for testing of truncated BCL11A variants. Both chips were constructed in an identical fashion as follows: a new CM5 chip was first docked into the instrument and primed in HBS-P+. Immobilisation was performed in manual run using parameters flow rate 10 µl/min; flow path 1,2,3,4; reference subtraction none. All surfaces were activated with a 420 s injection of NHS/EDC as described above. The flow path was changed to 2 and a 1/10 dilution of 0.19 mg/ml BCL11A in 10 mM sodium acetate pH 4.5 was injected for 930 s. The flow path was then changed to 3 and a 1/10 dilution of 0.34 mg/ml CHD8 was injected for 360 s. The flow path was then changed to 1,2,3,4 and the surface inactivated with a 420 s of 1 M Ethanolamine hydrochloride-NaOH pH 8.5. This gave final immobilisation levels of 5688RU and 5633RU for chip 1, respectively, and levels of 6048RU and 6146RU for chip 2, respectively.

CHD8 fragments were then analysed using chip 1, and BCL11A fragments using chip 2, in a multi-cycle kinetics format. Data was collected at a rate of 10 Hz using multi detection, with a sample compartment temperature of 4 °C, an analysis temperature of 25 °C, a flow rate of 30 µl/min and HBS-P+ as the running buffer. 10 startup cycles were used to allow for baseline stabilisation, which comprised 10 cycles of a 180 s injection of buffer followed by a 180 s dissociation time. Fragments were then injected at a single concentration as described in Table EV5 which comprised a neat injection of each purified protein. Cycles consisted of a 180 s association phase, a 180 s dissociation phase and a 60 s regeneration of 10 mM NaOH. A blank amine-coupled flow cell was used as a reference, and zero-concentration controls were used for double referencing. All analysis was carried out using BIAcore T200 evaluation software v1.0. Steady-state affinity models were then fitted to reference subtracted sensorgrams to calculate steady-state

$K_D$ values. Data used to generate SPR figures can be found in Dataset EV15–21 and Table EV6.

## SPR screening of in silico fragment hits

Fragments were purchased from Vitas-M Laboratory as powders and were resuspended to 20 mM in 100% DMSO. SPR was performed using a Biacore T200 instrument (Cytiva). For initial screening of the 8 purchased fragments, a new sensor chip CM5 (Cytiva, BR100399) was docked, and the instrument was primed in HBS-P+ buffer. Immobilisation of BCL11A to the chip surface was performed in a manual run, using parameters flowpath 1,2; flow rate 10 μl/min and no reference subtraction. The surface was then activated with a 420 s injection of NHS/EDC, as described above. After activation, the flow path was changed to 2 and a 1/10 dilution of 0.39 mg/ml BCL11A diluted in 10 mM sodium acetate pH 4 was injected for 1000 s. The flow path was then changed to 1,2 and the surfaces inactivated with a 420 s injection of ethanolamine, as described above, giving a final immobilisation level of 13000RU.

Fragments were then screened at a single high concentration in a multi-cycle kinetics format. Data was collected at a rate of 10 Hz using multi detection, with a sample compartment temperature of 25 °C, an analysis temperature of 25 °C, a flow rate of 30 μl/min and HBS-P+ with 2% DMSO as the running buffer. 10 startup cycles were used to allow for baseline stabilisation, which comprised 10 cycles of a 120 s injection of buffer followed by a 120 s dissociation time. Fragments were then injected at a concentration of 400 μM (diluted in HBS-P+ with 2% DMSO) and were injected for an association phase of 120 s and a dissociation phase of 120 s. No regeneration was performed. A blank amine-coupled flow cell was used as a reference, and zero-concentration controls were used for double referencing.

A follow-up 1:2 concentration series was then performed, ranging from 6.25 μM to 400 μM. For this follow-up experiment, a new series S CM5 sensor chip was docked into the instrument and primed in HBS-P+. Immobilisation was performed in manual run using parameters flow rate 10 μl/min, flow path 1,2,3,4; no reference subtraction. All surfaces were activated with a 420 s injection of NHS/EDC, as previously described, and the flow path was then changed to 2. A 1/10 dilution of 0.41 mg/ml BCL11A in 10 mM sodium acetate pH 4 was injected for 900 s. The flow path was then changed to 3 and a 1/20 injection of 0.70 mg/ml BCL11B was injected for 900 s. The flow path was then changed to 1,2,3,4 and the surfaces inactivated with a 420 s injection of ethanolamine, as described above, giving a final immobilisation level of 10265RU and 10747RU, respectively.

Fragments were then screened in a multi-cycle kinetics format. Data was collected at a rate of 10 Hz using multi detection, with a sample compartment temperature of 25 °C, an analysis temperature of 25 °C, a flow rate of 30 μl/min and HBS-P+ with 2% DMSO as the running buffer. Ten startup cycles were used to allow for baseline stabilisation, which comprised 10 cycles of a 300 s injection of buffer followed by a 300 s dissociation time. Fragments were then injected in a 1:2 dilution series ranging from 6.25 μM to 400 μM. Cycles comprised a 300 s association phase, a 300 s dissociation phase and a 30 s regeneration phase using 10 mM NaOH. A blank amine-coupled flow cell was used as a reference, and zero-concentration controls were used for double referencing. All analysis was carried out using BIAcore T200 evaluation software

v1.0. Steady-state affinity models were then fitted to reference subtracted sensorgrams to calculate steady-state $K_D$ values. Data used to generate SPR figures can be found in Dataset EV15–21 and Table EV6.

## SPR fragment inhibition assay

SPR was performed using a Biacore T200 instrument (Cytiva). The chip constructed in the "SPR analysis of the BCL11A-CHD8 interaction" section was used for this experiment. The chip was docked into the instrument and the system primed with HBS-P+ with 2% DMSO. Activity of the chip was confirmed by injection of 216 nM BCL11A, which gave a response of 213RU in Fc2-1. A single concentration of BCL11A was chosen for injection based on this response (54 nM, 1/80 dilution of 0.41 mg/ml BCL11A, to give a response of 100RU) and was injected either in the presence (400 μM of each) or absence of fragments. Fragments were added to BCL11A to give these final dilutions and were incubated at room temperature for 30 min.

BCL11A+/− fragments was then injected in an MCK format. Data was collected at a rate of 10 Hz using multi detection, with a sample compartment temperature of 25 °C, an analysis temperature of 25 °C, a flow rate of 30 μl/min and HBS-P+ with 2% DMSO as the running buffer. 10 startup cycles were used to allow for baseline stabilisation, which comprised 10 cycles of a 120 s injection of buffer followed by a 120 s dissociation time and a 60 s regeneration injection (20 mM NaOH). Samples were then injected with a 120 s association phase, a 120 s dissociation phase and a 60 s regeneration phase using 20 mM NaOH. A BSA-coupled flow cell was used as a reference, and zero-concentration controls were used for double referencing. All analysis was carried out using BIAcore T200 evaluation software v1.0. Steady-state affinity models were then fitted to reference subtracted sensorgrams to calculate steady-state $K_D$ values. Data used to generate SPR figures can be found in Dataset EV15–21 and Table EV6.

## Analysis of multi-OMICS data

### RNAseq

Initial quality control was performed on the raw FASTQ files using FastQC v0.11.8 and summarised using multiQC, version 1.9. To limit technical variation due to uneven sequencing depths, we subsampled, without replacement, raw samples with sequencing depth >40 M to 40 M reads (Mohorianu et al, 2017) resulting range of sequencing depths: 37.0 M to 40.0 M reads. The samples were aligned to the *M. musculus* genome [Ensemble 100.38] using STAR 2.7.0a (paired-end mode) (Dobin et al, 2013). Quantification was performed using featureCounts (Liao et al, 2014). The noise level across samples was assessed using noisyR (Moutsopoulos et al, 2021), applied to the count matrix, which was subsequently normalised using quantile normalisation (Bolstad et al, 2003). The visualisation and community-accessible overview were compiled using bulkAnalyseR (Moutsopoulos et al, 2023). Differential expression analyses were performed using edgeR (Robinson et al, 2010) and DESeq2 (Love et al, 2014); the enrichment analysis was performed using gprofiler2 (Raudvere et al, 2019) with all genes expressed above the signal/noise threshold as the background, all using Benjamini-Hochberg multiple testing correction. Geneset Enrichment Analysis (GSEA) was performed using the g:GOSt

functional profiling function of g:Profiler using all default options and the organism set to Mus musculus (mouse).

### ChIPseq

Initial quality control was performed on the raw FASTQ files using FastQC v0.11.8 and summarised using multiQC, version 1.9. To exclude high adaptor contamination, across all samples (determined using string pattern matching), the sequences were trimmed to 100 nt. To limit technical variation due to uneven sequencing depths we subsampled (without replacement) all samples, except the input 7c, to 50 M reads (Mohorianu et al, 2017). The samples were aligned to the *M. musculus* genome [Ensemble 100.38] using bowtie2 v2.4.2 (paired-end, local mode default parameters) (Langmead and Salzberg, 2012); next, narrow peaks were called using macs2 2.2.7.1 (Zhang et al, 2008) using the input 7c sample as control. Next, the coordinates of peaks found in at least one of the 6 samples were summarised (peaks matched if the midpoint of one peak fell within the span of the other peak). Using pyBigWig 0.3.17 (Ramirez et al, 2016), the corresponding amplitudes for these peaks were calculated (and scaled on the read length). An empirically determined signal/noise threshold of 20 was applied (Mohorianu et al, 2011). The peak expressions were scaled based on the mode expression of the corresponding control sample (control BCL11A IP or control CHD8 IP) to ensure comparable distributions across samples. To integrate with the mRNAseq analysis, we annotated/paired peaks with genes located within 5 kb flanking regions of the peaks. The visualisation and community-accessible overview were compiled using bulkAnalyseR (Moutsopoulos et al, 2023). Disappearing peaks in KDs compared to the corresponding control samples (e.g. CHD8 KD CHD8 IP vs control CHD8 IP) were identified using the $\log_2$FC threshold of 0.5. Disappearing peaks were then annotated by whether they coincided with exons, introns, promoter regions (defined as 2 kb region upstream from the TSS of a protein-coding gene), 3' and 5' UTRs, enhancers (obtained from Enhancer Atlas (Gao et al, 2016)), CpG islands (obtained from UCSC), transcription factor binding sites (obtained from JASPAR 2022 (Castro-Mondragon et al, 2022)) and CTCF sites (obtained from ENCODE (Consortium, 2012; Luo et al, 2020)); we focused on annotations where at least 50% of the peak overlapped with the feature. Motif analysis was performed for the disappearing peaks, identifying 6nt motifs and comparing them against all peaks and against regions upstream from genes as backgrounds. Fisher exact tests were focused on motif frequencies for the disappearing peak sets vs background sets; the motifs were labelled by significance level. Significant motifs ($p$-value < 0.05) were then matched against known TF motifs using TOMTOM (Gupta et al, 2007). A heatmap was created showing 2 kb upstream and downstream from the centre of disappearing peaks using deepTools (Ramirez et al, 2016), where samples were scaled so that the control samples in each case summed to the same abundance, for comparability.

### ATACseq

Initial quality control was performed on the raw FASTQ files using FastQC v0.11.8 and summarised using multiqc, version 1.9. To exclude high adaptor contamination from position 40 onwards we trimmed the reads to 40 nt. To limit technical variation due to uneven sequencing depths we subsampled (without replacement) all samples to 55 M reads (Mohorianu et al, 2017). The samples were aligned to the *M. musculus* genome [Ensemble 100.38] using

STAR 2.7.0a (paired-end mode), followed by peak calling using macs2 2.2.7.1. Next, the coordinates of peaks found in at least one of the 6 samples were summarised (peaks matched if the midpoint of one peak fell within the span of the other peak). Using pyBigWig 0.3.17 (Ramirez et al, 2016), the corresponding amplitudes for these peaks were calculated (and scaled on the read length). The peak expressions were denoised using noisyR (Moutsopoulos et al, 2021) applied to the count matrix and normalised using quantile normalisation (Bolstad et al, 2003).

To integrate with the RNAseq analysis, we annotated/linked peaks with protein-coding genes within 5 kb flanking regions of the peaks. The visualisation and community-accessible overview were compiled using bulkAnalyseR (Moutsopoulos et al, 2023).

## In silico fragment screening

A previously published crystal structure of DNA bound human BCL11A Znf domain (PDB: 6KI6) was used for structure based virtual screening (SBVS) (Yang et al, 2019). For the latter, a ligand search space (10 Å × 10 Å × 10 Å) was used around the C-terminal zinc finger domain (DNA was excluded) and GOLD suite version 5.3 (CCDC, Cambridge) was used with its default settings to screen three diversity chemical libraries namely VITAS Broadway (32,600 molecules), ChemBridge CORE (DIVERSet-CL, 50,000) and ChemDiv smart (55,000). For each case, the best 10% hits were retained after the VS and poses of each hit from each library were manually inspected. A list of the 20 best hits was generated based on scaffold diversity, predicted solubility and lack of any chemical alerts such as PAINS (judged via the SwissADME server). The top 8 hits were then purchased for experimental testing.

## 3D culture and colony assays

To study the effect of the fragments in vitro, 4T1, MDA-MB-231 and MCF-7 cells were grown to near confluency in a T75 flask and were detached in 0.05% Trypsin-EDTA (Gibco). Trypsin was inactivated through the addition of DMEM + 10% FBS and cells were pelleted at $500 \times g$ for 5 min at room temperature. For cell counting, the supernatant was removed, and the cell pellet was thoroughly resuspended in 10 ml PBS supplemented with 0.5 mg/ml Dispase (Sigma, D4693) and 0.01 mg/ml DNase (Sigma, D4513). The suspension was then pelleted at $500 \times g$ for 5 min at room temperature and spun down before being. Supernatant was removed and the pellet gently resuspended in 10 ml DMEM + 10% FBS and passed through a 30 μm cell strainer (CellTrics). Cells were counted using a haemocytometer.

4T1 cells were resuspended in Matrigel (Corning, 354230) on ice and seeded at a density of 250 cells/50 μl Matrigel domes into a pre-warmed 24-well plate. The Matrigel was left to solidify at 37 °C—upright for 10 min and subsequently upturned for 30 min. Once solidified, complete media (37 °C) was added to the wells. For the MDA-MB-231 and MCF-7 cells, a 30 μl layer of Matrigel was allowed to incubate (10 min at 37 °C) prior to seeding at a density of 2000 cells per 96-well. All cells were left to establish organoids (5 days for the MDA-MB-231 cells, and 3 days otherwise) in complete media, before being treated with complete media supplemented with in silico derived BCL11A-binding fragments at a final concentration of 200 μM. Treated media was replaced every second day. For 3D colony assays, treatments were repeated

in at least quadruplicate per experiment. Cells were incubated at 37 °C, 5% $CO_2$ for 6 days following treatment. Images were captured each day (EVOS, wide field 2x objective lens) for subsequent analysis on Image J.

### Image J analysis

Images were processed and analysed via Image J v1.54f. For each image, the scale was converted from pixels to micrometres ("Set Scale") based on the scalebar to allow future measurements to be recorded in micrometres. Multiple well images were combined using pairwise stitching (Preibisch et al, 2009). Images were then processed using "Enhance contrast" ($\sigma = 0.35$ and $0.5$ for 24 and 96 well images, respectively) and "Subtract background" (rolling = 50 for 96-well images) before detecting edges ("Find Edges"). A threshold was set to eliminate the background from the particle count (lower threshold = 35 and 20 for 24 and 96 well images, respectively, upper threshold = 255). 96 well images were converted to binary ("Convert to Mask), before all images were manually edited to remove debris and 2D colonies. Organoids with an area between 530 $\mu m^2$-infinity and a circularity between 0.05 and 1.00, were detected and counted ("Analyze Particles"). Organoids that were counted were outlined, generating an image of the organoid outlines superimposed onto the original. For each repeat, a summary of the count, total area, average organoid size, mean organoid size, average perimeter, average circularity, and average solidity was taken. Alongside this, the corresponding size, perimeter, circularity, and solidity were logged for all individual organoids/particles counted. Fold change in average colony size was performed relative to day 0 size and averaged across technical replicates ($n = 3$ or $4$). Fold changes in average colony size were fitted to a simple linear regression and analysed with $F$-tests in GraphPad Prism 10.

### EdU cell proliferation assay

4T1 cells were seeded at a density of 5e4 cells/12-well and left for 24 h prior to treatment with the fragments at a final concentration of 200 $\mu$M for a further 24 h. The cells were then treated with 10 $\mu$M EdU (Invitrogen, C10425) for 1 h 15 min at 37 °C, 5% $CO_2$ before harvesting as described previously. Cell pellets were fixed, stained, and washed as per the manufacturer's instructions. Cells were resuspended in 500 $\mu$l Click-iT saponin-based permabilisation and wash reagent (Invitrogen, C10425) and incubated in 1 $\mu$g/ml DAPI (Invitrogen, D1306) for 30 min. The cell suspension was analysed using a Sony cell sorter (SH800) and the percentage of cells per cell cycle phase was analysed using FlowJo (v10.10.0).

### Graphics

Graphics in Figs. 1A, 2A and 3C were created with BioRender.com.

### Statistical analysis

For statistical analysis of mRNA expression, an ordinary one-way ANOVA with a Dunnett multiple comparison correction was used. Means of each test condition were compared to the mean of the control condition. $P$ values reported are $*p < 0.05$; $**p < 0.01$; $***p < 0.001$ and $****p < 0.0001$.

For statistical analysis of colony number per 500 cells seeded, an ordinary one-way ANOVA with a Dunnett multiple comparison correction was used. Means of each test condition were compared

to the mean of the control condition. $P$ values reported are $*p < 0.05$; $**p < 0.01$; $***p < 0.001$ and $****p < 0.0001$.

For statistical analysis of xenograft tumour sizes, an ordinary one-way ANOVA with a Dunnett multiple comparison correction was used. Means of each test condition were compared to the mean of the control condition. $P$ values reported are $*p < 0.05$; $**p < 0.01$; $***p < 0.001$ and $****p < 0.0001$.

To statistically analyse the effect of small molecule fragments on breast cancer cell colonies, several algorithms and tests were applied. First, to assess fold change in 4T1, MDA-MB-231 or MCF7 colony size, lines were fitted with simple linear regression. This identified that the slopes of the growth rates differ between treatment groups for 4T1 cells ($n = 3$ passages, F = 3.999, $p = 0.025$) and MDA-MB-231 cells ($n = 3$ passages, F = 3.727, $p = 0.0226$) but not MCF7 cells ($n = 3$ passages, F = 0.5734, $p = 0.6368$). Second, statistics for cell cycle phase after fragment treatment were determined as follows: The G1 phase was analysed by Kruskal–Wallis test, whereas the G2/M and S phases were analysed by two-way ANOVA with post-hoc Dunnet's test identifying a significant difference between treatment with DMSO and Fragment 3 ($p = 0.0294$).

## Data availability

The authors declare that all data supporting the findings of this study and unprocessed images are available within the article and its supplementary information files or from the corresponding author upon reasonable request. The raw sequencing data is available on GEO (GSE174187). Processed data can be found and explored using https://bioinf.stemcells.cam.ac.uk/shiny/khaled_wUFt1bHfmC/waterhouse_allmodalities/.

The source data of this paper are collected in the following database record: biostudies:S-SCDT-10_1038-S44318-025-00447-8.

## Peer review information

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

## Acknowledgements

This study was primarily funded by a CRUK Drug Discovery Project (25850) and supported by Breast Cancer Now project grant (2017MayPR907), CRUK career establishment award (17348) and CRUK programme foundation award (DCRPGF\100010) to WTK. MFS and MM were supported by "Fondazione AIRC" (IG n. 21322), "Ministero della Salute", Italy (project RF-2019-12368937), NRRP- project Tech4You, NRRP - project ANTHEM, Sistema Integrato di Laboratori per L'Ambiente (SILA) PONa3_00341. We would like to thank staff members of the Anne McLaren Building (AMB) and other Cambridge animal facilities for facilitating mouse experiments. We would also like to thank the genomics core facilities at the Cambridge Stem Cell Institute (CSCI) and CRUK Cambridge Institute. We would also like to thank Dr Katherine Stott from the Biophysics facility at the Department of Biochemistry, University of Cambridge for assistance in performing SPR and other biophysical experiments. We would like to acknowledge the use of Biorender.com for the generation of some figure panels.

## Author contributions

**Mark Waterhouse**: Conceptualization; Data curation; Formal analysis; Validation; Investigation; Visualization; Methodology; Writing—original draft; Writing—review and editing. **Kyren Lazarus**: Data curation; Formal analysis; Investigation; Methodology. **Maria Francesca Santolla**: Data curation; Formal analysis; Investigation; Methodology. **Sara Pensa**: Resources; Data curation; Investigation; Methodology. **Eleanor Williams**: Data curation; Software; Formal analysis; Visualization; Methodology. **Abigail J Q Siu**: Data curation; Formal analysis; Investigation; Visualization; Methodology; Writing—review and editing. **Hisham Mohammed**: Resources; Data curation. **Irina Mohorianu**: Data curation; Software; Formal analysis; Investigation; Methodology. **Marcello Maggiolini**: Resources; Supervision. **Jason Carroll**: Resources; Supervision; Methodology. **Laura S Itzhaki**: Resources; Supervision; Funding acquisition; Methodology. **Taufiq Rahman**: Data curation; Software; Formal analysis; Supervision; Investigation; Visualization; Methodology. **Walid T Khaled**: Conceptualization; Formal analysis; Supervision; Funding acquisition; Investigation; Project administration; Writing—review and editing.

Source data underlying figure panels in this paper may have individual authorship assigned. Where available, figure panel/source data authorship is listed in the following database record: biostudies:S-SCDT-10_1038-S44318-025-00447-8.

## Disclosure and competing interests statement

The authors declare no competing interests.

