## [Peer Review File · The EMBO Journal]

CHD8 interacts with BCL11A to induce oncogenic transcription in triple negative breast cancer

Mark Waterhouse, Kyren Lazarus Maria Francesca Santolla, Sara Pensa, Eleanor Williams, Abigail J. Q. Siu, Hisham Mohammed, Irina Mohorianu, Marcello Maggolini, Jason Carroll, Laura S. Itzhaki, Taufiq Rahman, Walid T. Khaled

Corresponding author: Walid T. Khaled (wtk22@cam.ac.uk)

Review Timeline:

Submission Date:	6th Mar 24
Editorial Decision:	26th Apr 24
Revision Received:	20th Sep 24
Editorial Decision:	18th Nov 24
Revision Received:	10th Feb 25
Editorial Decision:	21st Mar 25
Revision Received:	24th Mar 25
Accepted:	8th Apr 25

Editor: Daniel Klimmeck

Transaction Report:

Dear Dr Khaled,

Thank you for submitting your manuscript for consideration by the EMBO Journal, as well as for your patience with our feedback at this time of the year. Your work has now been seen by two referees with expertise in cancer biology and transcription, whose comments are shown below.

Given the overall interest stated and broader angle of your findings, we are able to invite you to revise your manuscript experimentally to address the referees' comments. I need to stress though that we do require strong support from the referees on a revised version of the study in order to move on to publication of the work.

I would appreciate if you could contact me during the next weeks for exchange e.g. a video call to discuss your perspective on the comments and potential plan for revisions.

Please feel free to contact me if you have any questions or need further input on the referee comments.

When submitting your revised manuscript, please carefully review the instructions below.

Please feel free to approach me any time should you have additional questions related to this.

Thank you for the opportunity to consider your work for publication.

I look forward to your revision.

Best regards,

Daniel Klimmeck

Daniel Klimmeck, PhD
Senior Editor
The EMBO Journal

Instruction for the preparation of your revised manuscript:

- 1) a .docx formatted version of the manuscript text (including legends for main figures, EV figures and tables). Please make sure that the changes are highlighted to be clearly visible.
- 2) individual production quality figure files as .eps, .tif, .jpg (one file per figure).
- 3) a .docx formatted letter INCLUDING the reviewers' reports and your detailed point-by-point response to their comments. As part of the EMBO Press transparent editorial process, the point-by-point response is part of the Review Process File (RPF), which will be published alongside your paper.
- 4) a complete author checklist, which you can download from our author guidelines ([https://wol-prod-cdn.literatumonline.com/pb-assets/embo-site/Author Checklist%20-%20EMBO%20J-1561436015657.xlsx](https://wol-prod-cdn.literatumonline.com/pb-assets/embo-site/Author%20Checklist%20-%20EMBO%20J-1561436015657.xlsx)). Please insert information in the checklist that is also reflected in the manuscript. The completed author checklist will also be part of the RPF.
- 5) Please note that all corresponding authors are required to supply an ORCID ID for their name upon submission of a revised manuscript.
- 6) It is mandatory to include a 'Data Availability' section after the Materials and Methods. Before submitting your revision, primary datasets produced in this study need to be deposited in an appropriate public database, and the accession numbers and database listed under 'Data Availability'. Please remember to provide a reviewer password if the datasets are not yet public (see <https://www.embopress.org/page/journal/14602075/authorguide#datadeposition>). In case you have no data that requires deposition in a public database, please state so in this section. Note that the Data

Availability Section is restricted to new primary data that are part of this study.

7) Our journal encourages inclusion of *data citations in the reference list* to directly cite datasets that were re-used and obtained from public databases. Data citations in the article text are distinct from normal bibliographical citations and should directly link to the database records from which the data can be accessed. In the main text, data citations are formatted as follows: "Data ref: Smith et al, 2001" or "Data ref: NCBI Sequence Read Archive PRJNA342805, 2017". In the Reference list, data citations must be labeled with "[DATASET]". A data reference must provide the database name, accession number/identifiers and a resolvable link to the landing page from which the data can be accessed at the end of the reference. Further instructions are available at .

8) At EMBO Press we ask authors to provide source data for the main and EV figures. Our source data coordinator will contact you to discuss which figure panels we would need source data for and will also provide you with helpful tips on how to upload and organize the files.

Numerical data can be provided as individual .xls or .csv files (including a tab describing the data). For 'blots' or microscopy, uncropped images should be submitted (using a zip archive or a single pdf per main figure if multiple images need to be supplied for one panel). Additional information on source data and instruction on how to label the files are available at .

9) We replaced Supplementary Information with Expanded View (EV) Figures and Tables that are collapsible/expandable online (see examples in <https://www.embopress.org/doi/10.15252/emj.201695874>). A maximum of 5 EV Figures can be typeset. EV Figures should be cited as 'Figure EV1, Figure EV2' etc. in the text and their respective legends should be included in the main text after the legends of regular figures.

11) For data quantification: please specify the name of the statistical test used to generate error bars and P values, the number (n) of independent experiments (specify technical or biological replicates) underlying each data point and the test used to calculate p-values in each figure legend. The figure legends should contain a basic description of n, P and the test applied. Graphs must include a description of the bars and the error bars (s.d., s.e.m.).

We realize that it is difficult to revise to a specific deadline. In the interest of protecting the conceptual advance provided by the work, we recommend a revision within 3 months (25th Jul 2024). Please discuss the revision progress ahead of this time with the editor if you require more time to complete the revisions.

Referee #1:

In the manuscript "BCL11A interacts with CHD8 in TNBC to induce an oncogenic transcriptional programme" the authors investigate the molecular mechanisms through which the transcription factor BCL11A, which was previously identified as a driver of tumor progression in TNBC by the same authors, promotes tumorigenesis. They study the BCL11A tumor-specific protein-protein interaction (TSPPi) to identify target therapies affecting specifically cancer cells but not normal tissues. To do so, they analyze the BCL11A interactome in TNBC models and in murine B cells and identify the chromatin remodeler CHD8 as the unique interacting factor with BCL11A in all TNBC models but in B cells. They perform a battery of experiments to firmly demonstrate the physical interaction between these two proteins and the relevance of CHD8 for 4T1 tumorigenesis. Additional RNA-Seq and ChIP-seq experiments confirm these two factors share a common transcriptional and epigenetic program on a specific subset of genes involved in the cell cycle and DNA repair pathways. Then, the authors identify the physical interacting region between BCL11A and CHD8 by generating truncated mutants of both BCL11A and CHD8 and using SPR. Finally, they identify inhibitors of the BCL11A-CHD8 interaction and show some inhibitors that reduce TNBC colony formation capacity in 3D. The involvement of CHD8 in TNBC tumorigenesis is interesting and overall, the manuscript is easy to follow. The authors should address the generality of their findings in human models *in vivo*. Moreover, the figure legends are incomplete (see below). Addressing the following comments would make the article suitable for publication in the EMBO Journal:

Major Points

- 1) Figure 1: The authors performed the functional validation of the protumorigenic activity of CHD8 exclusively in murine 4T1 cells. Additional murine cell lines would help generalize their findings and the use of human TNBC cell lines (MDA-MB-231 or other lines) would enhance the human relevance of the findings. The authors should include a rescue with a CHD8 cDNA. They could also interrogate publicly available datasets (e.g., TCGA/METABRIC) to address whether BCL11A-CHD8 are highly expressed in TNBC compared with luminal tumors, clinical outcome prediction, correlation of expression between both proteins in TNBC clinical samples, etc. The authors show in Fig. 1B a WB of lysates from 4T1 cells used in the 3D colony assay; do the xenografted tumors also show a sufficient knockdown. Details on how tumor volumes were calculated are not indicated in the corresponding M&M section; they are needed because tumor volumes are very small.
- 2) Figure 3: The authors need to validate their findings from the RIME experiment also in the murine B cells by showing the absence of interaction between BCL11A and CHD8, to establish the interaction of these two proteins specifically in the TNBC cells. They could perform a co-IP of BCL11A-CHD8 in B-cells. It is an important negative control since they want to find a targeted therapy that affects only cancer cells.
- 3) Figure 4: In the fragment screening experiment, it seems that the 8 hits that prevent BCL11A-CHD8 complex formation *in vitro*, are active at very high concentrations. They are not really specific to BCL11A (Figure 4D-E), although the authors claim the opposite. The need to be used at this high concentration may elicit important off-target effects. Only fragments 3 and 8 are more specific to BCL11A than to BCL11B and the fold selectivity is between 3-5. Moreover, it seems that fragments 1, 4, and 8 are the most potent at reducing BCL11A binding to CHD8 (Figure 4G-H), but not the 3. So basically, the best candidate is fragment 8, which combines a good BCL11A binding inhibition (~60%) and a relatively good selectivity for BCL11A. Most of the other fragments are not specific or active.

This figure raises some concerns regarding the specificity and activity of the fragments that need to be studied more in-depth.

- 4) Figure 5: The authors conclude that fragments 3 and 5 are the only ones that reduce organoid growth (Fig. 5B-C). However, no statistics indicate the significance of the results (Fig. 5C-D). How do the authors explain that fragments 3 and 5, the two less potent fragments that block BCL11A-CHD8 binding (Fig. 4G-H), are the most effective ones in reducing organoid growth?

One could also conclude that, at least, fragment 4 increases organoid growth (Fig. 5C), which is weird as this is the second most potent fragment inhibiting BCL11A binding (Fig. 4G-H). This would invalidate the hypothesis that preventing BCL11A-CHD8 binding reduces tumor proliferation. How do the authors explain this?

Overall, Fig. 5 related to the inhibitory effect of the fragments identified in Figure 4 is not convincing: no statistics, no mention of technical replicate, no fragment concentration indicated... Additionally, the authors could have performed an IC50 or at least a drug range for each fragment to know at which concentration it may be active. What is the translational relevance if the fragment works in the millimolar range?

It seems that there is no correlation between the potency of blocking BCL11A-CHD8 interaction and the reduction in the proliferation of TNBC cells, which is problematic. Is the 3D organoid readout the most appropriate one? In Fig. 1, the authors used a sphere-forming assay for example.

The qPCR data are not convincing, because fragment 3 which reduces the proliferation of TNBC cells (Fig. 5C) does not have any effect at the transcriptional level. How do the authors explain this?

The authors could also do Co-IP experiments to test whether these fragments are potent at reducing BCL11A-CHD8 interaction in cultured cells and not only using SPR assays.

As the first aim of the manuscript is to identify targeted therapies specific for cancer cells, the use of non-transformed cells (e.g., murine B cells) is needed to test the effect of these fragments on normal cells. It could be that these fragments that block BCL11A affect not only the binding of CHD8 but the binding of other proteins to BCL11A in other tissues.

CHD8 is a chromatin remodeller with a genuine helicase ATPase enzymatic activity (hence potentially "druggable"), contrarily to BCL11A which is a transcription factor. Wouldn't it be better to target CHD8 itself rather than the interaction with BCL11A? Is there any CHD8 inhibitor that the authors could test on TNBC cells versus B cells and/or ER+ breast cancer cells to also show the subtype-specific killing effect?

Figure legends

- In the legend section, the authors often do not properly describe the figures but interpret the results. Example: Fig. 1C: The legend is a pure interpretation of the panel, but does not describe the figure.
- The authors do not mention the statistical tests they used nor what the stars correspond to. The error bars are also not described. It's neither in the legends nor in the method section which is problematic for claiming an effect. Also, some graphs lack statistics while they are interpreted in a way that requires proper tests. An obvious example is Fig 5C.
- Some key experimental details are also missing in many panels. Example Fig. 5C: what is the concentration of the fragments used for the 3D colony assay?
- The number of technical and experimental replicates is not indicated.
- Some panels in the legends are not fitting: for example, Fig. 2 at the very end "(C) ATAC seq analysis..." while we are after panel E description.
- Some figure legends are also lacking description, like in Fig. 1: how was the protein name motif panel generated? Does the size of the protein name correspond to its statistical significance? Or enrichment score? Wouldn't it be clearer to display a small table with some values? Does the color of the protein name refer to something?
- Some panels do not have axis titles, nor protein names for western blots... This is a recurrent issue across the paper. If graphs are in different panels within the same figure, they need proper naming.
- The font of the axis is also not appropriate and should be harmonized across the manuscript: for example, in Fig. 1E, it is impossible to read anything while in Fig. 4B, the font size is 30.
- The gene and protein nomenclature is also problematic across the whole paper. Like Figure 1 panel B, CHD8 should be written *Chd8* as the authors refer to the murine gene.

Minor Points

1) Figure 1: The authors could also add FACS data with PI/annexin staining or Ki67 to confirm that CHD8 impinges proliferation because this effect is mostly based on functional annotation of the RNA-ChIP Seq data.

Use the same type of graph for both box plots from Fig. 1C and 1D.

2) Suppl. Fig. 3: The authors could add the hallmark functional annotation to confirm that BCL11A-CHD8 transcriptional program is associated with proliferation.

3) How did the authors define the cut-off for selecting the BCL11A interacting proteins in all models? Couldn't they define a cut-off to obtain a shorter list? having 1500 proteins pulled down seems gigantic and difficult to interpret.

Referee #2:

This is a potentially interesting study to identify proteins that might selectively interact with BCL11A in triple negative breast cancer, and, therefore, might provide a potential interface for selective drug targeting. The authors identified the chromatin remodeler CHD8 and demonstrate that it plays a role in TNBC and interacts with BCL11A to regulate gene expression. The authors identified a fragment that inhibited the interaction and showed that the fragment reduced colony formation of mouse 4T1 cells in vitro. In general, these studies were carefully performed, but the authors fail to discuss significant limitations of their approach as detailed below. The following major concerns need to be addressed.

- The authors nicely showed that CHD8 did not interact with BCL11A in B cells but whether this interaction exists in the brain was not tested. In the Introduction, the authors mentioned that BCL11A is expressed in brain cells. CHD8 is also expressed in the brain, specifically oligodendrocytes, and mutation/haploinsufficiency of CHD8 has been associated with autistic-like neurodevelopmental effects. Therefore, it is important to determine whether this interaction is important for brain function before the interaction with BCL11A can be validated as a therapeutic target.
- CHD8 has been co-purified with the MLL histone modifying complex so it is likely that a much larger protein super complex may be involved with BCL11A. How does this affect the design of fragments.
- The co-IP experiments do not prove direct protein-protein interactions. The SPR analyses of purified proteins do demonstrate direct interactions in vitro. What about in vivo using a proximity ligation assay.

- The authors use 4T1 cells for functional studies. Unlike the conventional 3T3 colony forming assays these assays for epithelial tumors don't always correlate with tumorigenicity in vivo, so the authors should be careful about overinterpreting the significance of these results. In their in vivo studies with the 4T1 model did they analyze effects on lung metastases that would be clinically more relevant than primary tumor growth.
- In Figure 5D, Fragment 5 downregulated only 2 of the genes significantly, and Fragment 3 seemed to show a reverse effect. These results are not strong enough to validate that the reduced colony formation was driven by inhibition of the interaction.
- In Figure 1D, the authors never indicated the control for this experiment.
- Since the authors have used PDX models as well as the claudin-low MDA-MB231 and 4T1 cells, do they know in which TNBC subtype the BCL11A and CHD8 are expressed.

Referee #1:

In the manuscript "BCL11A interacts with CHD8 in TNBC to induce an oncogenic transcriptional programme" the authors investigate the molecular mechanisms through which the transcription factor BCL11A, which was previously identified as a driver of tumor progression in TNBC by the same authors, promotes tumorigenesis. They study the BCL11A tumor-specific protein-protein interaction (TSPPI) to identify target therapies affecting specifically cancer cells but not normal tissues. To do so, they analyze the BCL11A interactome in TNBC models and in murine B cells and identify the chromatin remodeler CHD8 as the unique interacting factor with BCL11A in all TNBC models but in B cells. They perform a battery of experiments to firmly demonstrate the physical interaction between these two proteins and the relevance of CHD8 for 4T1 tumorigenesis. Additional RNA-Seq and ChIP-seq experiments confirm these two factors share a common transcriptional and epigenetic program on a specific subset of genes involved in the cell cycle and DNA repair pathways. Then, the authors identify the physical interacting region between BCL11A and CHD8 by generating truncated mutants of both BCL11A and CHD8 and using SPR. Finally, they identify inhibitors of the BCL11A-CHD8 interaction and show some inhibitors that reduce TNBC colony formation capacity in 3D. The involvement of CHD8 in TNBC tumorigenesis is interesting and overall, the manuscript is easy to follow. The authors should address the generality of their findings in human models *in vivo*. Moreover, the figure legends are incomplete (see below). Addressing the following comments would make the article suitable for publication in the EMBO Journal:

We would like to thank the reviewer to taking the time to examine our manuscript. We appreciate the reviewer's positive feedback and have outlined our response to their comments below.

Major Points

1) Figure 1: The authors performed the functional validation of the protumorigenic activity of CHD8 exclusively in murine 4T1 cells. Additional murine cell lines would help generalize their findings and the use of human TNBC cell lines (MDA-MB-231 or other lines) would enhance the human relevance of the findings. The authors should include a rescue with a CHD8 cDNA.

We thank the reviewer for the suggestion. We have performed a similar shRNA-based experiment in MDA-MB-231 which demonstrate comparable results to those seen in 4T1 cells in terms of reduced colony formation. We have not performed a CHD8 cDNA rescue experiment, but we have performed an orthogonal validation approach using Crispr/Cas9 and 2 independent gRNAs to KO Chd8 in 4T1 cells. These results further highlight the role of CHD8 in tumorigenesis. We include these results in the revised manuscript (FIGURE 1b and Supp Figure 1a-c) and are included here as well (Figure R1)

A**B****C***Figure R1*

They could also interrogate publicly available datasets (e.g., TCGA/METABRIC) to address whether BCL11A-CHD8 are highly expressed in TNBC compared with luminal tumors, clinical outcome prediction, correlation of expression between both proteins in TNBC clinical samples, etc.

We thank the reviewer for this suggestion. We have investigated the expression of BCL11A in METABRIC and TCGA in our previous publication (Khaled et al. 2015) and we also investigated the co-expression with CHD8 which we now include in Supp figure 1 and Figure R2 below.

E

Figure R2

The authors show in Fig. 1B a WB of lysates from 4T1 cells used in the 3D colony assay; do the xenografted tumors also show a sufficient knockdown. Details on how tumor volumes were calculated are not indicated in the corresponding M&M section; they are needed because tumor volumes are very small.

We thank the reviewer for identifying this missing information. We have updated the materials and methods accordingly.

2) Figure 3: The authors need to validate their findings from the RIME experiment also in the murine B cells by showing the absence of interaction between BCL11A and CHD8, to establish the interaction of these two proteins specifically in the TNBC cells. They could perform a co-IP of BCL11A-CHD8 in B-cells. It is an important negative control since they want to find a targeted therapy that affects only cancer cells.

We thank the reviewer for this suggestion. We have performed Co-IP experiments using spleen samples from mice, which are rich in B cells. We show that BCL11A pull down in the spleen brings down only BCL11A but not CHD8 despite it being present in the cell. This further validates the RIME results and we have now included this new data in the revised manuscript (Figure 3a and Figure R3 below).

Figure R3

3) Figure 4: In the fragment screening experiment, it seems that the 8 hits that prevent BCL11A-CHD8 complex formation in vitro, are active at very high concentrations. They are not really specific to BCL11A (Figure 4D-E), although the authors claim the opposite. The need to be used at this high concentration may elicit important off-target effects.

We thank the reviewer for this comment. This is a common feature of all fragment screens which are intended to be the first step towards a more developed compound. Initial fragment hits generally have weak binding affinity to their targets (uM-mM range) and in the future medicinal chemistry would be needed to optimise their affinity. This level of compound development is expensive and requires working closely with medicinal chemists, all of which is beyond the scope of the current study, which is focusing on the initial discovery of the cancer specific protein-protein interactions and the identification of the lead fragments. Although we see significant differences in affinity of the fragments between BCL11A and BCL11B, this could be improved further at the next stage of medicinal chemistry. We will add more about this in the discussion so its clearer for the reader.

Only fragments 3 and 8 are more specific to BCL11A than to BCL11B and the fold selectivity is between 3-5. Moreover, it seems that fragments 1, 4, and 8 are the most potent at reducing BCL11A binding to CHD8 (Figure 4G-H), but not the 3. So basically, the best candidate is fragment 8, which combines a good BCL11A binding inhibition (~60%) and a relatively good selectivity for BCL11A. Most of the other fragments are not specific or active. This figure raises some concerns regarding the specificity and activity of the fragments that need to be studied more in-depth.

We appreciate the reviewer's point on selectivity versus activity in the SPR assay. In addition, it is important to note that the inhibitory activity shown in the in vitro SPR assay may not necessarily translate to inhibition in a cell-based assay as there might be other factors involved (eg, cell permeability, allosteric effects, other members of the protein complex). The SPR assay here is primarily to show that inhibition of the interaction is possible, and we then subsequently test whether any of these putative inhibitors are active in cells. However, we have now repeated the drug treatment on two human breast cancer cell lines, MCF7 and MDA-MB-231. MDA-MB-231 is a TNBC cell line that expresses high levels of BCL11A and CHD8 responded in a similar way to 4T1 (Figure 5D and Figure R4). In contrast, MCF7 which is a Luminal cell line that does not express high levels of BCL11A was not affected by any of the fragments (Figure 5D and Figure R4). In addition, we have also performed CO-IP experiments on Bcl11a and Chd8 using 4T1 cells in presence or absence of fragments and show that they indeed disrupt the Bcl11a-Chd8 interaction in cells (Figure 5C and Figure R4). Collectively these results suggest that off-target effects are not in play and further confirm the preference of the identified fragments in disrupting the BCL11A-CHD8 interactions in TNBC but not Luminal breast cancer.

Figure R4

4) Figure 5: The authors conclude that fragments 3 and 5 are the only ones that reduce organoid growth (Fig. 5B-C). However, no statistics indicate the significance of the results (Fig. 5C-D). How do the authors explain that fragments 3 and 5, the two less potent fragments that block BCL11A-CHD8 binding (Fig. 4G-H), are the most effective ones in reducing organoid growth? One could also conclude that, at least, fragment 4 increases organoid growth (Fig. 5C), which is weird as this is the second most potent fragment inhibiting BCL11A binding (Fig. 4G-H). This would invalidate the hypothesis that preventing BCL11A-CHD8 binding reduces tumor proliferation. How do the authors explain this? Overall, Fig. 5 related to the inhibitory effect of the fragments identified in Figure 4 is not convincing: no statistics, no mention of technical replicate, no fragment concentration

indicated... Additionally, the authors could have performed an IC50 or at least a drug range for each fragment to know at which concentration it may be active. What is the translational relevance if the fragment works in the millimolar range?

We thank the reviewer for the comment. As highlighted in the previous point it is important to note that the inhibitory activity shown in the in vitro SPR assay may not necessarily translate to inhibition in a cell-based assay as there might be other factors involved (eg, cell permeability, allosteric effects, other members of the protein complex). The SPR assay here is primarily to show that inhibition of the interaction is possible, and we then subsequently test whether any of these putative inhibitors are active in cells. These experiments were repeated on three separate occasions, with at least three replicates per fragment per experiment (n>9 total). We have now included the statistical analysis in the revised manuscript along with a better description of the experimental methods. Fragments were used at a single concentration of 200uM, which was selected as this was the highest possible dose allowed without inducing cell death due to the presence of DMSO. The used fragment concentration is included in the methods, but we will update the figure for clarity.

In terms of translation, these fragments are starting points for further development into higher affinity and higher potency chemical entities. That is why we have not performed any IC50 experiments as we are aware that the fragments will need further development by medicinal chemist due which is beyond the scope of this study. During this chemical evolution, it is expected that affinity, potency and selectivity would all be greatly increased, providing a window into clinical translation.

It seems that there is no correlation between the potency of blocking BCL11A-CHD8 interaction and the reduction in the proliferation of TNBC cells, which is problematic. Is the 3D organoid readout the most appropriate one? In Fig. 1, the authors used a sphere-forming assay for example.

We agree with the reviewer on the importance of selecting the correct model. The sphere-forming assays described in Figure 1 are the same assays being used in Figure 5. We have counted colonies but see no significant difference for these fragments at their current potency. We believe that enhancing potency through chemical derivatisation will result in a more robust anti-tumour response, leading to tumour arrest and death. However, we have now repeated the drug treatment on two human breast cancer cell lines, MCF7 and MDA-MB-231. MDA-MB-231 is a TNBC cell line that expresses high levels of BCL11A and CHD8 responded in a similar way to 4T1 (Figure 5d and Figure R4). In contrast, MCF7 which is a Luminal cell line that does not express high levels of BCL11A was not affected by any of the fragments (Figure 5d and Figure R4). In addition, we have also performed CO-IP experiments on Bcl11a and Chd8 using 4T1 cells in presence or absence of fragments and show that they indeed disrupt the Bcl11a-Chd8 interaction in cells (Figure 5c and Figure R4).

The qPCR data are not convincing, because fragment 3 which reduces the proliferation of TNBC cells (Fig. 5C) does not have any effect at the transcriptional level. How do the authors explain this?

This can again be explained through potency. The ability of these fragments currently to induce the full effects seen in shRNA-based knockdown coupled with RNAseq is unlikely to occur until optimisation of the fragments has been performed. However, the reduction in colony growth rate and the perturbation of several BCL11A-CHD8 controlled gene targets, especially by fragment 5, gives confidence and provides justification for this optimisation.

The authors could also do Co-IP experiments to test whether these fragments are potent at reducing BCL11A-CHD8 interaction in cultured cells and not only using SPR assays. As the first aim of the manuscript is to identify targeted therapies specific for cancer cells, the

use of non-transformed cells (e.g., murine B cells) is needed to test the effect of these fragments on normal cells. It could be that these fragments that block BCL11A affect not only the binding of CHD8 but the binding of other proteins to BCL11A in other tissues. CHD8 is a chromatin remodeller with a genuine helicase ATPase enzymatic activity (hence potentially "druggable"), contrarily to BCL11A which is a transcription factor. Wouldn't it be better to target CHD8 itself rather than the interaction with BCL11A? Is there any CHD8 inhibitor that the authors could test on TNBC cells versus B cells and/or ER+ breast cancer cells to also show the subtype-specific killing effect?

We thank the reviewer for the suggestion. We have now repeated the drug treatment on two human breast cancer cell lines, MCF7 and MDA-MB-231. MDA-MB-231 which is a TNBC cell line expresses high levels of BCL11A and CHD8 responded in a similar way to 4T1 (Figure 5d and Figure R4). In contrast, MCF7 which is a Luminal cell line that does not express high levels of BCL11A was not affected by any of the fragments (Figure 5d and Figure R4). In addition, we have also performed CO-IP experiments on Bcl11a and Chd8 using 4T1 cells in presence or absence of fragments and show that they indeed disrupt the Bcl11a-Chd8 interaction in cells (Figure 5c and Figure R4). CHD8 alone does not present itself a good target for inhibition (See TCGA and METABRIC data supp figure 1/Figure R2). CHD8 is ubiquitously expressed and is also a well-known negative regulator of beta catenin. Inhibiting helicase activity will produce effects in all tissues and increase activation of beta catenin, which may increase the risk of cancer in various tissues and create a toxicology profile that is too high risk to pursue this option. BCL11A and inhibiting a tumour specific protein-protein interaction (TSPPI) therefore presents itself as an attractive alternative option in this scenario.

Figure legends

- In the legend section, the authors often do not properly describe the figures but interpret the results. Example: Fig. 1C: The legend is a pure interpretation of the panel, but does not describe the figure.
- The authors do not mention the statistical tests they used nor what the stars correspond to. The error bars are also not described. It's neither in the legends nor in the method section which is problematic for claiming an effect. Also, some graphs lack statistics while they are interpreted in a way that requires proper tests. An obvious example is Fig 5C.
- Some key experimental details are also missing in many panels. Example Fig. 5C: what is the concentration of the fragments used for the 3D colony assay?
- The number of technical and experimental replicates is not indicated.
- Some panels in the legends are not fitting: for example, Fig. 2 at the very end "(C) ATAC seq analysis..." while we are after panel E description.
- Some figure legends are also lacking description, like in Fig. 1: how was the protein name motif panel generated? Does the size of the protein name correspond to its statistical significance? Or enrichment score? Wouldn't it be clearer to display a small table with some values? Does the color of the protein name refer to something?
- Some panels do not have axis titles, nor protein names for western blots... This is a recurrent issue across the paper. If graphs are in different panels within the same figure, they need proper naming.
- The font of the axis is also not appropriate and should be harmonized across the manuscript: for example, in Fig. 1E, it is impossible to read anything while in Fig. 4B, the font size is 30.
- The gene and protein nomenclature is also problematic across the whole paper. Like Figure 1 panel B, CHD8 should be written Chd8 as the authors refer to the murine gene.

We would like to thank the reviewer for pointing out these improvements and inconsistencies. We have updated the manuscript accordingly, including:

- *Updating figure legends to have greater detail*

- *Inclusion of statistics and definitions of significance*
- *Updating some figure panels for greater clarity*
- *Inclusion of replicates*
- *Figures checked for consistency, axis titles added where necessary*
- *Gene and protein names corrected as appropriate*

Minor Points
 1) Figure 1: The authors could also add FACS data with PI/annexin staining or Ki67 to confirm that CHD8 impinges proliferation because this effect is mostly based on functional annotation of the RNA-ChIP Seq data.

Use the same type of graph for both box plots from Fig. 1C and 1D.

2) Suppl. Fig. 3: The authors could add the hallmark functional annotation to confirm that BCL11A-CHD8 transcriptional program is associated with proliferation.

Thank you for this suggestion. We will add this to Supp Fig. 3

3) How did the authors define the cut-off for selecting the BCL11A interacting proteins in all models? Couldn't they define a cut-off to obtain a shorter list? having 1500 proteins pulled down seems gigantic and difficult to interpret.

We thank the authors for their comments and understand that these are very big datasets. To identify true BCL11A interacting proteins, we first removed proteins that appeared in both the BCL11A IP and control (IgG) IP lists. We then only considered proteins in the BCL11A IP list that had at least two unique peptides identified in the mass spec analysis. We will clarify this in the methods section.

Referee #2:

This is a potentially interesting study to identify proteins that might selectively interact with BCL11A in triple negative breast cancer, and, therefore, might provide a potential interface for selective drug targeting. The authors identified the chromatin remodeler CHD8 and demonstrate that it plays a role in TNBC and interacts with BCL11A to regulate gene expression. The authors identified a fragment that inhibited the interaction and showed that the fragment reduced colony formation of mouse 4T1 cells in vitro. In general, these studies were carefully performed, but the authors fail to discuss significant limitations of their approach as detailed below. The following major concerns need to be addressed.

We thank the reviewer for taking the time to review our manuscript and for their positive feedback. We have outlined our response to their comments below.

- The authors nicely showed that CHD8 did not interact with BCL11A in B cells but whether this interaction exists in the brain was not tested. In the Introduction, the authors mentioned that BCL11A is expressed in brain cells. CHD8 is also expressed in the brain, specifically oligodendrocytes, and mutation/haploinsufficiency of CHD8 has been associated with autistic-like neurodevelopmental effects. Therefore, it is important to determine whether this interaction is important for brain function before the interaction with BCL11A can be validated as a therapeutic target.

We thank the reviewer for this suggestion. We now include the results from a RIME experiment performed on WT female Brain tissue and we can indeed pull down BCL11A but not CHD8. This supports the specific nature of the BCL11A-CHD8 interaction in TNBC. (Supplementary Table 20).

- CHD8 has been co-purified with the MLL histone modifying complex so it is likely that a much larger protein super complex may be involved with BCL11A. How does this affect the design of fragments.

We agree with the reviewer that BCL11A and/or CHD8 may also engage with “super complexes” and indeed RIME suggest there are other partners involved in this complex. However, the fragments designed here are based on a published BCL11A zinc finger motif which is predicted not to be involved in protein-protein interactions.

- The co-IP experiments do not prove direct protein-protein interactions. The SPR analyses of purified proteins do demonstrate direct interactions in vitro. What about in vivo using a proximity ligation assay.

While we appreciate the utility of proximity ligation assay, we feel that this would not differ much from the results obtained from Co-IP. Proximity ligation assays require distances less than 40nm (4×10^{11} angstroms). At the protein-protein interaction level, this is a very large distance and would not provide additional information about the “directness” of the interaction. We feel that the combination of Co-IP and SPR is sufficient to prove the interaction. In addition, we have now included an additional experiment where we have also performed Co-IP on Bcl11a and Chd8 using 4T1 cells in presence or absence of fragments and show that they indeed disrupt the Bcl11a-Chd8 interaction in vivo (Figure 5c and Figure R4).

- The authors use 4T1 cells for functional studies. Unlike the conventional 3T3 colony forming assays these assays for epithelial tumors don't always correlate with tumorigenicity in vivo, so the authors should be careful about overinterpreting the significance of these results. In their in vivo studies with the 4T1 model did they analyze effects on lung metastases that would be clinically more relevant than primary tumor growth.

We thank the reviewer for their comment. We have performed a similar shRNA-based experiment in MDA-MB-231 which demonstrate comparable results to those seen in 4T1 cells in terms of reduced colony formation. In addition, we have performed an orthogonal validation approach using Crispr/Cas9 and 2 independent gRNAs to KO Chd8 in 4T1 cells. These results further highlight the role of CHD8 in tumorigenesis. We include these results in the revised manuscript (FIGURE 1b and Supp Figure 1a-c) and are pasted here as well (Figure R1). We have not investigated the effect on lung metastasis which we believe is beyond the scope of the current focus of this study characterising the BCL11A-CHD8 TSPPI.

- In Figure 5D, Fragment 5 downregulated only 2 of the genes significantly, and Fragment 3 seemed to show a reverse effect. These results are not strong enough to validate that the reduced colony formation was driven by inhibition of the interaction.

We thank the reviewer for this comment and believe that this phenomenon can be explained through potency. The ability of these fragments currently to induce the full effects seen in shRNA-based knockdown coupled with RNAseq is unlikely to occur until optimisation of the fragments has been performed. This is a common feature of all fragment screens which are intended to be the first step towards a more developed compound. Initial fragment hits generally have weak binding affinity to their targets (μM - mM range) and in the future medicinal chemistry would be needed to optimise their affinity. This level of compound development is expensive and requires working closely with medicinal chemists, all of which

is beyond the scope of the current study, which is focusing on the initial discovery of the cancer specific protein-protein interactions and the identification of the lead fragments.

- In Figure 1D, the authors never indicated the control for this experiment.

Thank you for identifying this missing information. We will update the manuscript accordingly. The controls were C57/B6 WT mammary tissue.

- Since the authors have used PDX models as well as the claudin-low MDA-MB231 and 4T1 cells, do they know in which TNBC subtype the BCL11A and CHD8 are expressed.

BCL11A is predominantly expressed in Integrative cluster 10 which tend to have loss of Chromosome 5q and gain of Chromosome 9p and 10p which are also patients that tend to have poor 5-year survival (Russnes et al 2017 The American journal of pathology). Chd8 on the other hand is not associated with any of the integrative clusters, further supporting the importance of the TSPPI.

Dear Dr Khaled,

Thank you for submitting your revised manuscript (EMBOJ-2024-117193R) to The EMBO Journal, as well for your patience with our response, which got delayed due to protracted referee input. Your amended study was sent back to the referees for their scientific re-evaluation, and we have received detailed comments from both of them, which I enclose below. As you will see, the experts state that the work has been substantially improved by the revisions and they are supportive of the work, pending satisfactory revision.

The referees state that additional complementation is required to comprehensively support a case for the BCL11A-CHD8 interaction as a pathophysiological event relevant to TNBC. They give concrete support on how to bolster these claims and consolidate the findings by additional experiments.

We have discussed the input in the editorial team and agree the biological relevance of the molecular interaction is a key aspect of the work, hence the points by the referee #1 should be considered. We thus invite you for a final minor revision of the work.

We also need you to take care of a number of issues related to formatting and data presentation as detailed below, which should be addressed at re-submission.

Please contact me at any time if you have additional questions related to the referees' arguments and below points.

Thank you for giving us the chance to consider your manuscript for The EMBO Journal. I look forward to your final revision.

Best regards,

Daniel Klimmeck

>> Please add maximally five keywords to your study.

>> Author Contributions: Please remove the author contributions information from the manuscript text. Note that CRediT has replaced the traditional author contributions section as of now because it offers a systematic machine-readable author contributions format that allows for more effective research assessment. and use the free text boxes beneath each contributing author's name to add specific details on the author's contribution.

More information is available in our guide to authors.
<https://www.embopress.org/page/journal/14602075/authorguide>

>> Funding: enter the following funding information in the list of funders in our online system: Breast Cancer Now project grant (2017MayPR907), "Fondazione AIRC" (IG n. 21322), "Ministero della Salute", Italy (project RF-2019-12368937), NRRP- project Tech4You, NRRP - project ANTHEM, Sistema Integrato di Laboratori per L'Ambiente (SILA) PONA3_00341.

>> Section order should be corrected: title page with complete author information, abstract, keywords, introduction, results, discussion, methods, data availability section, acknowledgements, disclosure and competing interests statement, references, main figure legends, tables, expanded figure legends.

>> Appendix file: Title page with the subtitle Appendix for BCL11A interacts with CHD8 in TNBC to induce an oncogenic transcriptional programme and ToC and page numbers missing; Appendix Figure legends should be removed from manuscript

file, remaining only in Appendix PDF, but placed below the corresponding figures; nomenclature should be Appendix Figure S1-S11 throughout the manuscript and Appendix PDF instead of Supp Fig 1-11 on figures.

>> Add a Reagents and Tools table to the Methods section, listing key reagents, experimental models, software and relevant equipment.

>> Dataset EV legends: tables Supplementary Table 1-13, 20-21, 23-28 should be renamed to Dataset EV1-EV21 with the corresponding callouts and legends uploaded as separate tabs in each Excel file.

>> Add a separate 'Statistical Analysis' section to the Methods part, detailing the algorithms and statistical tests applied.

>> Data availability section: Remove the referee token and ensure the GSE174187 dataset is publicly available. Add the specific URL.

>> Tables Supplementary Table 14-19 and 22 should be renamed to Table EV1-EV7 with the corresponding callouts and legends uploaded in each Excel file.

>> Author Checklist: third column (pink boxes) needs to be completed with the section names when the response is positive (Yes).

>> Please provide source data for the study as to the separate request e-mail by my colleague Hannah Sonntag.

>> As to our journal policies we kindly ask you to check & clarify i) re-use of Western blot data without citation in the figure legend. (Figure 1 B&E - Appendix Fig S1D). ii) Figure 3C diagram of cells appears to also be used in the published manuscript: Front Cell Neurosci. 2023 Jan 19; 17:1082180, Accession ID: PMC9893793 , PMID: 36744004 , Licensed: CC BY. Is this a stock footage used?

>> Consider additional changes and comments from our production team as indicated below:

- Data citations: no comments
- Figure Legends (main + EV):

1. Please note that the exact p values are not provided in the legends of figures 1b, e; 5e.
2. Please indicate the statistical test used for data analysis in the legends of figures 5b, d.
3. Please note that in figure 1e; there is a mismatch between the annotated p values in the figure legend and the annotated p values in the figure file that should be corrected.
4. Please note that the box plots need to be defined in terms of minima, maxima, centre, bounds of box and whiskers, and percentile in the legends of figures 1b-c.
5. Please note that the box plot needs to be defined in terms of minima, maxima, centre, bounds of box, and percentile in the legend of figure 1d.
6. Please note that information related to n is missing in the legends of figures 1b, d-e.
7. Although 'n' is provided, please describe the nature of entity for 'n' in the legend of figure 5e.
8. Please note that the error bars are not defined in the legends of figures 1b, e; 5e.
9. Please note that for heatmap present in figure 2c; a numbered scale bar is not provided. This needs to be rectified.

Referee #1:

In the revised version of the manuscript "BCL11A interacts with CHD8 in TNBC to induce an oncogenic transcriptional programme", the authors have addressed some of the points that were raised in the initial review. The new data showing absence of interaction between BCL11A and CHD8 in B cells or in the brain are very convincing, and the new Co-IP experiments showing disruption of the interaction suggest that the selected fragments disrupt the interactions in a TNBC cell line. The figure legends were improved, although still contain some interpretation of the data. The points below still need clarification.

Major points:

1) Is the in vitro knockdown/knockout of CHD8 kept in vivo? The authors should show a Western Blot of CHD8 specifically in the xenografted tumors. In addition, details on how tumor volumes were calculated (e.g. the formula used for the calculations) are not indicated in the corresponding M&M section; these details are needed because tumor volumes from Figure 1B, middle panel (4T1 knockdown xenograft tumors), are extremely small, especially compared to the left panel (4T1 CRISPR KO xenograft tumors).

2) The authors claim that their study lays ground for future optimization of the identified fragments to develop a peptide-based therapy. While it is understandable that optimization of the fragments is beyond the scope of the study, the experimental evidence that supports these claims remain weak. Essentially, it all comes down to this one experiment presented in Figure 5A, B and D, which shows differences in 3D colony size. First, the authors should indicate on the graphs the statistical significance (asterisks or p-values). They use linear regression to statistically compare the slope of the curves but what is the p-value referring to? For which contrast? Which fragments are being compared? All the fragments have the same statistical significance and p-values? In the Figure 5B, the legend says "the slopes of the growth rates differ between treatment groups", but which groups? In what way do they differ? Why is there only a single p-value for a graph that includes seven experimental conditions? Furthermore, some fragments seem to even accelerate 3D growth in 4T1 cells, is that significant (e.g. fragment 4)?

To be confident that this is a real biological effect, the authors should perform non-parametric t-test or ANOVA at day 6 of treatment and see whether the effect of the fragment is statistically significant or not. The CoIP data (Figure 5C) convincingly show disruption of CHD8-BCL11A interaction upon treatment with the fragments, therefore one should expect a clear difference in organoids growth, if this interaction is essential to TNBC biology. Also, the terminology "growth rate" or "colony size" is rather vague, do the authors mean reduced proliferation or cytotoxic effect? This is not discussed.

The authors could use a second readout for the effect of the fragments such as a sphere assay, as shown in Figure 1 or a 2D proliferation assay, or a 2D colony-forming assay. Ki-67 staining to check whether there is reduced proliferation when treating with fragment 3 or 5 would also be an appropriate readout. For now, it is not even clear which fragments significantly decrease 3D organoids growth.

Also, from the pictures of the wells shown below the graphs one can hardly see any effect. Finally, the qPCR data provide weak evidence that disrupting the interaction has any biological effect in altering the gene expression program governed by CHD8-BCL11A interaction, although the interaction is clearly disrupted (CoIP data). Fragment 3 has basically no effect although it is the most efficient at reducing proliferation. Isn't there a better readout?

In conclusion, the authors should strengthen the validation part (point 2), before the manuscript is suitable for publication in a journal like EMBO.

Referee #2:

The authors have addressed some of our major concerns by including several additional experiments. The overall translational significance for TNBC is still questionable.

Referee

#1:

In the revised version of the manuscript "BCL11A interacts with CHD8 in TNBC to induce an oncogenic transcriptional programme", the authors have addressed some of the points that were raised in the initial review. The new data showing absence of interaction between BCL11A and CHD8 in B cells or in the brain are very convincing, and the new Co-IP experiments showing disruption of the interaction suggest that the selected fragments disrupt the interactions in a TNBC cell line. The figure legends were improved, although still contain some interpretation of the data. The points below still need clarification.

We thank the reviewer for their comments and are very grateful for the time and effort spent on reviewing our manuscript. We are pleased that the B-cell and brain RIME alongside the new Co-IP experiments have been positively received. We also thank the reviewer for their latest set of comments and suggestions, which we have answered below:

Major points:

1) Is the in vitro knockdown/knockout of CDH8 kept in vivo? The authors should show a Western Blot of CHD8 specifically in the xenografted tumors. In addition, details on how tumor volumes were calculated (e.g. the formula used for the calculations) are not indicated in the corresponding M&M section; these details are needed because tumor volumes from Figure 1B, middle panel (4T1 knockdown xenograft tumors), are extremely small, especially compared to the left panel (4T1 CRISPR KO xenograft tumors).

Details for how tumours are measured have been added to the "Mouse Experiments" section of the methods and can be found below:

"Animals were monitored for tumour growth and tumour volume was measured every 48-72 hours using calipers. All tumours were collected, processed, and stored in an identical manner. Briefly, tumours that exceeded 1.2 cm² were excised from humanely killed mice and were snap frozen on dry ice for isolation of protein and RNA. Tumours were only observed in the mammary epithelium in the Brca1f/f; p53+/-;Blg-Cre mice"

The western blots are of the cells in culture not the xenograft tumours. The tumour reduction is quite significant we would struggle to extract protein from the small outgrowth.

2) The authors claim that their study lays ground for future optimization of the identified fragments to develop a peptide-based therapy. While it is understandable that optimization of

the fragments is beyond the scope of the study, the experimental evidence that supports these claims remain weak. Essentially, it all comes down to this one experiment presented in Figure 5A, B and D, which shows differences in 3D colony size. First, the authors should indicate on the graphs the statistical significance (asterisks or p-values). They use linear regression to statistically compare the slope of the curves but what is the p-value referring to? For which contrast? Which fragments are being compared? All the fragments have the same statistical significance and p-values? In the Figure 5B, the legend says "the slopes of the growth rates differ between treatment groups", but which groups? In what way do they differ? Why is there only a single p-value for a graph that includes seven experimental conditions? Furthermore, some fragments seem to even accelerate 3D growth in 4T1 cells, is that significant (e.g. fragment 4)? To be confident that this is a real biological effect, the authors should perform non-parametric t-test or ANOVA at day 6 of treatment and see whether the effect of the fragment is statistically significant or not.

We have spoken to a statistician and we performed a linear regression to represent the change in fold change in size over a period of time, which we refer to as the rate of growth. This looked to compare the global effects of the fragments within each cell line, and therefore the single p-value represents the fact that there are significant differences between the treatment groups within 4T1 cells and MDA-MB-231s (TNBC cells), but not in MCF7s (luminal breast cancer cells). We have now clarified the wording within the figure legend (please see below) and we thank the reviewer for their comments. As the reviewer suggested we also performed an ANOVA at day 6 of treatment which was not significant. We now include more evidence that the fragments (which are not drugs) not only disrupt the biophysical interaction but also impact the biology of the cells when measured using **three** independent methods 3D colony size, Co-IP disruption and EdU proliferation assay (see below for more detail). We are not claiming these are final drugs that should be tested on TNBC patients but the combination of all these findings warrant the future development of these fragments into potent inhibitors of the BCL11A-CHD8 PPI.

(B) Fold change in average 4T1 colony size relative to day 0 size upon treatment of fragments at 200uM across 6 days of treatment, fitted with simple linear regression identifying that the slopes of fold change in average colony size over time differ globally within 4T1s (n=3 passages, F=3.999, p=0.025) (left panel).

(E) Fold change in average colony size relative to day 0 size comparing human cell lines MCF-7 and MDA-MB-231 upon treatment of fragments 1, 3 and 5 at 200uM. 3D colony assay images visualising the phenotypic changes in MCF-7 and MDA-MB-231 colony size. The scale bars represent 1000um (left panel) Fitted simple linear regressions to fold change in average colony size over time identifies global differences in MDA-MB-231 cells (n=3 passages, F = 3.727, p = 0.0226), with no global difference in the slope (n=3, F = 0.5817, p = 0.6319) or intercept (n=3 passages, F = 0.5734, p = 0.6368) in MCF-7 cells (right panel).

The CoIP data (Figure 5C) convincingly show disruption of CDH8-BCL11A interaction upon treatment with the fragments, therefore one should expect a clear difference in organoids growth, if this interaction is essential to TNBC biology. Also, the terminology "growth rate" or "colony size" is rather vague, do the authors mean reduced proliferation or cytotoxic effect? This is not discussed.

The authors could use a second readout for the effect of the fragments such as a sphere assay, as shown in Figure 1 or a 2D proliferation assay, or a 2D colony-forming assay. Ki-67 staining to check whether there is reduced proliferation when treating with fragment 3 or 5 would also be an appropriate readout. For now, it is not even clear which fragments significantly decrease 3D organoids growth.

We thank the reviewer for their comments on Figure 5 and agree that further experimental evidence would support the effect of the fragments on 3D colony size, as such we have furthered the analysis to include EdU proliferation assays to demonstrate that the effect on colony growth rate is due to reduced proliferation.

Fig. 5D - Percentage of 4T1 cells in G1, S and G2/M phase following treatment of fragments 1, 3 and 5 at 200uM for 24h, determined by flow cytometry. Data presented as median and range (n=3 passages), G1 analysed by Kruskal-Wallis test, G2/M and S phase analysed by two-way ANOVA with post-hoc Dunnett's test identifying a significant difference between treatment with DMSO and Fragment 3 ($p=0.0294$).

We appreciate that the CoIP data convincingly show the disruption of CHD8-BCL11A interaction upon the treatment, we think that this, in combination with our new proliferation data supports the fact that the phenotypic reductions in colony size are biologically relevant.

Also, from the pictures of the wells shown below the graphs one can hardly see any effect. Finally, the qPCR data provide weak evidence that disrupting the interaction has any biological effect in altering the gene expression program governed by CHD8-BCL11A interaction, although the interaction is clearly disrupted (CoIP data). Fragment 3 has basically no effect although it is the most efficient at reducing proliferation. Isn't there a better readout?

We appreciate that it can be hard to observe differences from the pictures of the wells as they have been displayed. We also agree that it is hard to draw strong conclusions from the

qPCR data that has been included, likely as the fragments we are using to perturb this interaction are very weak affinity. As a result, we have made major changes to figure 5 that are detailed below:

- We have removed all of the well pictures from the original Figure 5 and moved them to a new Appendix Figure S12. The images displayed here are from the day 6 time point only, as we felt that at the 0 day time point the colonies were too small and did not add any useful information. The day 6 images are displayed in a much bigger size, and we believe make the difference between DMSO/non-active fragments and the active fragments 3 and 5 much clearer. Please see the new Appendix Figure S12 on the following page:

Appendix Figure S12 Representative brightfield images comparing effect of fragments on colony size.

3D colony assay images visualising the phenotypic changes in (A) 4T1 colony size following 6 days of treatment with all 6 binders at 200uM, (B) MDA-MB-231 and MCF7 colony size following 6 days of treatment with 3 selected binders at 200uM. All scale bars represent 2000um.

- For the main Figure 5, we have instead only included larger pictures of the active fragments 3 and 5 alongside one inactive fragment (fragment 1) and the DMSO control for comparison. Again, we believe this more clearly demonstrates the activity of fragments 3 and 5. Please see the new panels below:

Fig 5B - Fold change in average 4T1 colony size relative to day 0 size upon treatment of fragments at 200uM across 6 days of treatment, fitted with simple linear regression identifying that the slopes of the growth rates differ between treatment groups (n=3 passages, $F=3.999$, $p=0.025$) (left panel). 3D colony assay images visualising the phenotypic changes in 4T1 colony size following 6 days of treatment with the 3 selected binders. The scale bar represents 2000um (right panel).

Fig 5E Fold change in average colony size relative to day 0 size comparing human cell lines MCF-7 and MDA-MB-231 upon treatment of fragments 1, 3 and 5 at 200uM. 3D colony assay images visualising the phenotypic changes in MCF-7 and MDA-MB-231 colony size. The scale bars represent 1000um (left panel). Fitted simple linear regressions to the growth rates identify differences in intercepts in the slopes of MDA-MB-231 cells (n=3 passages, $F = 3.727$, $p = 0.0226$), with no difference in the slope (n=3, $F = 0.5817$, $p = 0.6319$) and intercept (n=3 passages, $F = 0.5734$, $p = 0.6368$) of MCF-7 cells upon treatment (right panel).

- We have removed the qPCR and instead used a more biologically relevant EdU cell proliferation assay to understand the effects of fragments on the proliferation of TNBC cells. This shows a significant decrease in the proportion of 4T1 cells progressing into S phase when treated with fragment 3, and a non-significant decrease with fragment 5, versus DMSO control. We see no changes in cell cycle phase from the non-active fragment 1. We believe this much more clearly demonstrates a biological effect of these fragments. This is very much in line with what we would expect to see based on our RNAseq data, which demonstrates that BCL11A and CHD8 positively regulate a large shared set of cell cycle-related genes/proteins. Please see the results below:

Fig. 5D - Percentage of 4T1 cells in G1, S and G2/M phase following treatment of fragments 1, 3 and 5 at 200uM for 24h, determined by flow cytometry. Data presented as median and range (n=3 passages), G1 analysed by Kruskal-Wallis test, G2/M and S phase analysed by two-way ANOVA with post-hoc Dunnet's test identifying a significant difference between treatment with DMSO and Fragment 3 ($p=0.0294$).

In conclusion, the authors should strengthen the validation part (point 2), before the manuscript is suitable for publication in a journal like EMBO.

We again thank the reviewer as we believe we have greatly strengthened our manuscript by addressing these concerns. In summary, we have identified fragment inhibitors of the BCL11A-CHD8 interaction that are able to block the interaction in both biophysical SPR-based assays as well as CoIP assays. These fragments reduce the growth rate of TNBC cells but not other types of breast cancer cells (MCF-7). They are also able to reduce the proliferative capacity of TNBC cells by perturbing progression of the cell cycle into S-phase. We believe that these fragments now represent very promising starting points for

derivatisation into more potent inhibitors of this complex.

Referee #2:

The authors have addressed some of our major concerns by including several additional experiments. the overall translational significance for TNBC is still questionable.

We thank the reviewer for their comments and the ongoing support in reviewing our manuscript. We would urge the reviewer to consider our newer data showing that our fragment inhibitors of the BCL11A-CHD8 interaction are able to perturb cell cycle by reducing progression of TNBC cells into S-phase, versus non-active fragments or DMSO. Please see these new results below.

Fig. 5D - Percentage of 4T1 cells in G1, S and G2/M phase following treatment of fragments 1, 3 and 5 at 200uM for 24h, determined by flow cytometry. Data presented as median and range (n=3 passages), G1 analysed by Kruskal-Wallis test, G2/M and S phase analysed by two-way ANOVA with post-hoc Dunnet's test identifying a significant difference between treatment with DMSO and Fragment 3 ($p=0.0294$).

We believe this much more clearly demonstrates a biological effect of these fragments. This is very much in line with what we would expect to see based on our RNAseq data, which demonstrates that BCL11A and CHD8 positively regulate a large shared set of cell cycle-related genes/proteins. We believe that these fragments now represent very promising starting points for derivatisation into more potent inhibitors of this complex.

Dear Dr Khaled,

Thank you for submitting your revised manuscript (EMBOJ-2024-117193R1) to The EMBO Journal, as well for your patience with our response. Your amended study was sent back to the referee #1 for his-her scientific reassessment, however this expert was at this time not able to evaluate your amended work. In the interest of time, we have now editorially reassessed your response to the final critique and found the issues raised to be addressed satisfactorily.

Thus, we are pleased to inform you that your manuscript has been accepted in principle for publication in The EMBO Journal.

We still need you to take care of a minor remaining issue related to data annotation as detailed below, which should be addressed at final re-submission.

Also, as you might have noted from our webpage, every paper at the EMBO Journal now includes a 'Synopsis', displayed on the html and freely accessible to all readers. The synopsis includes a 'model' figure as well as 2-5 one-short-sentence bullet points that summarize the article. I would appreciate if you could provide this figure and the bullet points.

Thank you for giving us the chance to consider your manuscript for The EMBO Journal. I look forward to your final revision.

Again, please contact me at any time if you need any help or have further questions.

Best regards,

Daniel Klimmeck

>> Author Checklist: the third column (pink boxes) needs to be completed with the section names when the response is positive (Yes).

The authors addressed the remaining editorial issues.

Dear Dr Khaled,

Thank you for submitting the revised version of your manuscript. I have now evaluated your amended manuscript and concluded that the remaining minor concerns have been sufficiently addressed.

I am thus pleased to inform you that your manuscript has been accepted for publication in the EMBO Journal.

Related, I would like to hereby ask your consent on keeping the referee response figures included in this file.

On a different note, I would like to alert you that EMBO Press offers a format for a video-synopsis of work published with us, which essentially is a short, author-generated film explaining the core findings in hand drawings, and, as we believe, can be very useful to increase visibility of the work. Please see the following link for representative examples and their integration into the article web page:

<https://www.embopress.org/doi/full/10.15252/embo.2019103932>

Best regards,

Daniel Klimmeck

Daniel Klimmeck, PhD
Senior Editor
The EMBO Journal
EMBO
Postfach 1022-40
Meyerohofstrasse 1
D-69117 Heidelberg
contact@embojournal.org